# Measuring Meta-Cultural Competency: A Spectral Framework for LLM Knowledge Structures

## Abstract

Most cultural evaluation frameworks for Large Language Models (LLMs) compare model outputs with ground-truth answers, capturing mainly factual awareness. This overlooks whether models internalize broader cultural structures and pluralism. In this paper, we introduce a spectral-analysis-based framework to uncover large-scale structural patterns in models' cultural knowledge. We test eight LLMs of different sizes across nine cultural domains (food, religion, language, etc.) spanning 170 countries, comparing their learned structures with human data. Results show that instruction-tuned LLMs align more closely with human patterns than older models like GPT-2 and GPT-J. However, model size is not always an advantage, and performance asymptotes: Llama-8B and Gemma-2B perform as well or better than their larger-sized counterparts: Llama-70B and Gemma-9B. These findings differ from model rankings on existing probing-based cultural benchmarks, showing that our method captures a distinct aspect of cultural competency. Furthermore, initial simulation-based experiments demonstrate that compared to traditional metrics of cultural awareness, the proposed spectral metric is better able to predict a model's ability to serve a user from an unfamiliar background.

## 1 Introduction

> *"And so these men of Indostan disputed loud and long, each in his own opinion exceeding stiff and strong. Though each was partly in the right, and all were in the wrong!"* - Saxe (1871).

The parable of the blind men and the elephant illustrates the limitations of partial observation: each observer perceives something correct in isolation, yet none captures the larger whole. Current evaluation frameworks for LLMs exhibit a similar limitation in the cultural domain. Most approaches assess cultural awareness by comparing model outputs to ground-truth answers within narrow, localized contexts (Nadeem et al., 2021; Yin et al., 2022; Jha et al., 2023). While informative, current benchmarks primarily evaluate a model's cultural knowledge through discrete facts, such as recognizing that *Biryani* is a common food in India or that Indian and Pakistani cuisines share notable similarities. These are *microstructural properties* of cultural knowledge. However, such benchmarks fail to capture the *macrostructure*, that is, the broader, recurring patterns, such as how many overarching categories of cuisines exist, whether they are hierarchically structured or randomly distributed, and how much variation occurs within and across national cuisines (Sorensen et al., 2024; Strauss, 1992). This broader capacity, often referred to as *variational awareness*, constitutes a critical dimension of *meta-cultural competency* (Leung et al., 2013; Sharifian, 2013), the innate human ability to recognize, interpret, and navigate cultural variation across levels of familiarity (Noshadi & Dabbagh, 2015). It is essential for making sense of both familiar and unfamiliar cultural contexts.

Unlike databases, which store and enable the retrieval of point facts, LLMs are powerful compression engines that structure knowledge in ways that allow them to generate appropriate responses across diverse cultural domains and settings (Talmor et al., 2020; Geva et al., 2021; Pan et al., 2025). Hence, analyzing macrostructures provides a more holistic measure of *cultural awareness*, or as we shall argue, of *variational awareness*, and eventually of *meta-cultural competency*. Macrostructural evaluation also has pragmatic advantages: global structural patterns are easier and more reliable

to elicit from humans than idiosyncratic preferences (Triandis, 1989; Shweder, 1991; Matsumoto, 2007). For instance, some domains, such as currency, are highly country-specific, while others, like house numbering,[1] are more cross-nationally uniform. Testing whether LLMs capture such relative patterns directly probes their ability to model cultural structure and variation, a ground truth that is tractable and human-verifiable. Yet, despite this promise, macrostructures remain underexplored in LLM evaluation.

We address this gap by introducing a spectral-analysis-based framework (Klema & Laub, 1980; Wall et al., 2003; Stoica et al., 2005; Abdi, 2007) for evaluating the macrostructures of LLM cultural knowledge. Using this framework, we study eight models of different sizes across nine cultural domains (food, religion, language, currency, holidays, etc.) spanning 170 countries, and show how macrostructural evaluation complements microstructural benchmarks.

Overall, **our contributions** are as follows: (i) **We introduce a spectral-analysis-based framework** that shifts cultural evaluation from probing local microstructures to modeling domain-level macrostructures, which provides a principled way of evaluating variational awareness, a necessary first step towards meta-culturally competent AI. (ii) Using data from 170 countries and nine cultural domains, **we provide the first large-scale macrostructural analysis** of eight LLMs of different sizes, and show that macrostructural evaluation is tractable, interpretable, and aligns with human expectations. (iii) Finally, **simulation-based experiments** demonstrate that our spectral metric predicts a model's ability to serve users from unfamiliar backgrounds better than existing probing-based benchmarks, providing a way to measure a system's *explication strategy* (Sharifian, 2013), which is a higher-order competency expected from meta-cultural AI systems in user-facing scenarios.

## 2  BACKGROUND AND RELATED WORK

### 2.1  NECESSARY DEFINITIONS

**Cultural Consensus Theory (CCT):** Weller (2007) describes CCT (Romney et al., 1987) as *"a collection of analytical techniques and models that can be used to estimate cultural beliefs and the degree to which individuals know or report those beliefs"*. It infers shared knowledge by analyzing patterns of agreement across responses, assuming that when a common cultural model exists, disagreement reflects variation in knowledge rather than truth. Using factor analysis of the agreement matrix, CCT identifies the dominant consensus through the first eigenvalue and derives individual competence scores from factor loadings. In essence, CCT provides a way to recover both collective knowledge and individual reliability without requiring predefined ground truth. We use CCT as a guiding framework for designing our macrostructure evaluation metrics.

**Microstructures and Macrostructures:** These are complementary levels of knowledge, where microstructures capture localized factual associations (e.g., "the currency of Japan is Yen"), while macrostructures reflect the global organization of knowledge across domains (e.g., how currencies distribute across countries or how cuisines cluster geographically). This distinction mirrors other fields, such as Physics, where microstates are particle-level configurations and macrostates are emergent system properties (Reif, 2009; Pathria, 2017; Huang, 2008). In cognitive science, higher-level structures enable reasoning beyond local facts (Carey, 2000; Lake et al., 2017; Kemp & Tenenbaum, 2008); and in anthropology, recurring cultural patterns emerge from localized practices (Lévi-Strauss, 1963). As Anderson (1972) argued, "more is different", where higher levels of complexity yield properties irreducible to micro-level descriptions. Hence, evaluating both levels is essential, since microstructures provide factual building blocks, whereas macrostructures reveal systemic cultural patterns.

**Meta-Cultural Competency:** Having defined how knowledge is structured (micro/macro), we now discuss **meta-cultural competency**, an innate human ability to navigate cultural knowledge (Noshadi & Dabbagh, 2015), and how it motivates our evaluation framework. Meta-cultural competency enables us to communicate and negotiate cultural conceptualizations across contexts (Sharifian, 2013). It comprises three components: (i) *Variational awareness*: The meta-knowledge that cultural practices, beliefs, and preferences vary across groups, which provides the foundation for recognizing the limits of one's own knowledge. (ii) *Explication strategies*: The ability to act on this

---

[1] House numbering conventions may vary, but the numerals themselves are mostly universal.

awareness by asking clarifying questions or making one's own assumptions explicit in unfamiliar settings. (iii) *Negotiation strategies*: The conversational moves that help interlocutors resolve misinterpretations and jointly construct meaning. For instance, a traveler in Japan who notices unfamiliar dining customs may recognize (variational awareness) that their prior assumptions about mealtime behavior do not hold, ask whether it is customary to say *"Itadakimasu"* before eating (explication), and adapt their own behavior after clarification (negotiation).

Saha et al. (2025) argues that meta-cultural competency is a necessary system-level ability for cross-cultural AI, such that models can detect when they are out of distribution (via variational awareness) and adapt by seeking or incorporating missing knowledge (through explication and negotiation). We extend this framework and use macrostructures, which measure the system-level distributions of knowledge, as a lens to measure variation awareness. Finally, through a practical user-facing experiment, we show that evaluating macrostructures indeed predicts models' meta-cultural competency.

## 2.2 RELATED WORK

Knowledge estimation from LLMs has been researched primarily in two directions: (i) **Response-based**, where carefully crafted queries are utilized to elicit factual or commonsense knowledge from the models, which are evaluated against curated datasets containing ground-truths(Chern et al., 2023; Sun et al., 2024; Wang et al., 2020; Petroni et al., 2019; Jiang et al., 2021; Newman et al., 2021; Jiang et al., 2020; Nguyen et al., 2023; Wu et al., 2025). Most existing works in evaluating LLMs' cultural competence are response-based. (Nadeem et al., 2021; Nangia et al., 2020; Wan et al., 2023; Jha et al., 2023; Li et al., 2024; Cao et al., 2023; Tanmay et al., 2023; Rao et al., 2023; Kovač et al., 2023). Some methods (Kharchenko et al., 2024; LI et al., 2024; Dawson et al., 2024) also analyze the model-generated responses along theoretical frameworks such as Hofstede's cultural dimensions (Hofstede, 2001; Geert & Hofstede, 2004) and measure their proximity with cultures, where high proximity indicates better value alignment between the nearby cultures and the values portrayed by the model's response. (ii) **Internals-based**, where approaches leverage the LLM attention map (Wang et al., 2020), activation function (Burns et al., 2022), or model parameters (Kazemnejad et al., 2023) to decide the suitability of the information extracted from the LLMs. Many works have also studied spectral analysis (Staats et al., 2024; Mukherjee et al., 2009) and found it useful in improving LLMs (Saha et al., 2024; Hartford et al., 2024; Sharma et al., 2023). There has also been an interesting line of research in quantifying uncertainty in LLM prediction (Huang et al., 2024; Liu et al., 2024b; Ma et al., 2025; Ye et al., 2024). While such approaches provide an intuitive view of model capabilities, they are inherently limited by the sensitivity of the models in the prompt and the decoding strategies employed. Although advancements have been made in the mechanistic interpretability of LLMs (Conmy et al., 2023; Nanda et al., 2023) in general, there is a lack of applying such internals-based methods for evaluating LLMs' cultural alignment (Yu et al., 2025).

It is also prudent to distinguish macrostructural analysis from model calibration, which evaluates whether probabilistic predictions reflect true likelihoods (Niculescu-Mizil & Caruana, 2005; Guo et al., 2017). Calibration operates at the microstructural level and often degrades under domain shift, leaving deeper class (in the long tail of culture) and domain-specific miscalibrations unresolved (Kull et al., 2019; Saha et al., 2025). In contrast, our macrostructural approach evaluates whether models encode coherent, domain-level properties (e.g., connectedness of cuisines vs. disconnected currencies), shifting the focus from local accuracy to global knowledge organization.

Cultural benchmarks, which exhaustively capture real-world diversity, are difficult to construct. Many such datasets (Wang et al., 2024; Rao et al., 2024; Myung et al., 2024; Zhou et al., 2024; Putri et al., 2024; Mostafazadeh Davani et al., 2024; Wibowo et al., 2024; Owen et al., 2024; Chiu et al., 2024; Liu et al., 2024a; Koto et al., 2024b) fail to capture the pluralism of human values and preferences, limiting their effectiveness for measuring cultural competence (Sorensen et al., 2024).

## 3 APPROACH

If we conceptualize LLMs' encoded knowledge as structured networks, where nodes represent concepts and edges capture associations between them (Tenenbaum et al., 2011; Nickel & Kiela, 2017), microstructural evaluations can be thought of as probing these networks locally by querying the model with prompts of varying compositionality (e.g., first-order, second-order), effectively analyz-

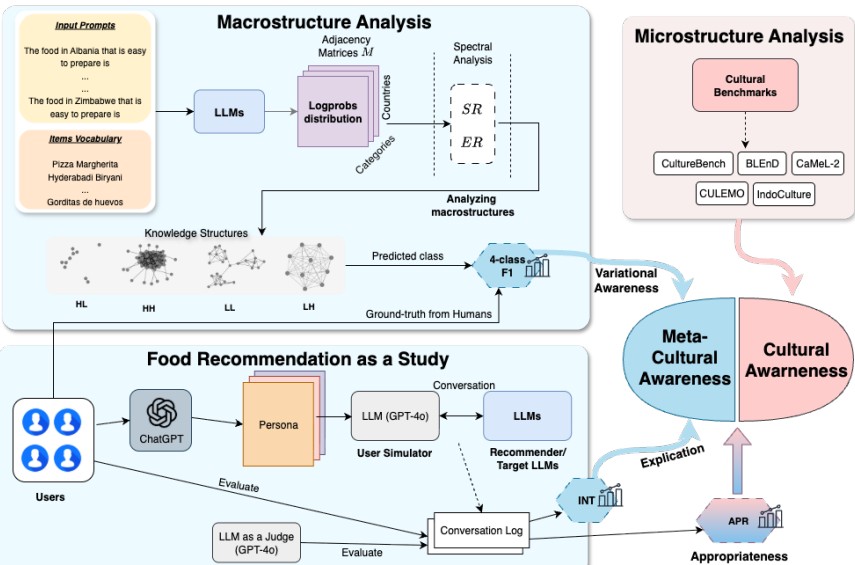

Figure 1: Overview of our end-to-end evaluation framework. We assess meta-cultural competency by analyzing macrostructures for variational awareness (top left) and explication in recipe recommendation (bottom left), validated through downstream appropriateness (bottom right), and compared against microstructural benchmarks (top right).

ing small subgraphs. For instance, asking a model to list the local dishes of a region (first-order), or to combine multiple regions (n-th order), and validating against curated lists, which reveals only pointwise associations in the subgraphs. While informative, such evaluations remain limited to microstructures and do not capture whether the model has internalized broader organizational principles, such as how cuisines cluster across regions or how domains like food, religion, and currency interrelate, as reflected in their macrostructural patterns.

Macrostructures, by summarizing distributions of cultural knowledge across domains, provide a natural lens for evaluating variational awareness in LLMs. This perspective shifts evaluation from isolated factual queries to understanding how the model organizes and relates knowledge across domains. We next present a formal definition of our framework and the ensuing experiments. Later, we extend the analysis to a simulation-based setting that tests whether models operationalize their variational awareness in user-facing interactions, thereby exhibiting explication strategies and, more generally, meta-cultural competence as reflected in the appropriateness of their responses.

## 3.1 FORMAL DEFINITION

Let $D = \{d_1, ..., d_m\}$ be a set of $m$ cultural domains and $C = \{c_1, ..., c_n\}$ be a set of $n$ cultural proxies, which in our case are countries, since countries are the most common and well defined demographic proxy, unlike most other demographic variables (Adilazuarda et al., 2024). Each domain $d_i$ contains $k$ semantically similar questions $Q^{d_i} = \{q_1, ..., q_k\}$ posed over a set of $t$ items $I^{d_i} = \{i_1, ..., i_t\}$. For example, in the currency domain, $I^{d_i}$ could be an exhaustive list of all world currencies. Let $M_\theta$ be a model parameterized by $\theta$. For each country $c \in C$, the model answers a domain-specific question $q_j \in Q^{d_i}$ with a probability distribution $p_j^c$ over the $t$ items, where $\sum p_j^c = 1$. This distribution represents the model's microstructural knowledge: its probabilistic associations between countries and items. Most cultural evaluations compare $p_j^c$ with ground-truth distributions. We extend this to capture macrostructural knowledge.

For a domain $d_i$, we collect country-level distributions across all $n$ countries to form an $n \times t$ matrix $H^{d_i}$. From this, we derive an $n \times n$ adjacency matrix $A^{d_i}$, where each entry encodes pairwise cosine similarity of country distributions:

$$A^{d_i} = D^{-1} H H^\top D^{-1}, D = \mathrm{diag}(\|p_j^{c_1}\|_2, ..., \|p_j^{c_n}\|_2); A^{d_i}, D \in \mathbb{R}^{n \times n} \tag{1}$$

We analyze $A^{d_i}$ using two spectral metrics: (i) The entropy-based **Effective Rank (ER)**, which measures the number of significant dimensions of $A^{d_i}$ (Roy & Vetterli, 2007). ER = 1 when dominated by one eigenvalue, and ER = $n$ when eigenvalues are uniform. (ii) Inspired by CCT, **Spectral Gap Ratio (SR)** captures the gap between the first and second eigenvalues. A high SR indicates strong cross-country similarity, while a low SR reflects country-specific variation. Let $\{\lambda_1, ..., \lambda_n\}$ denote the set of all eigenvalues of $A^{d_i}$ ordered in descending order.

$$\text{ER} = \exp(-\sum_{i=1}^{n} \tilde{\lambda}_i \log \tilde{\lambda}_i), \text{where } \tilde{\lambda}_i = \frac{\lambda_i}{\sum_{j=1}^{n} \lambda_j}, \lambda_i > 0; \text{SR} = \frac{\lambda_1}{\lambda_2} \tag{2}$$

### 3.1.1 Categorizing Macrostructures of Cultural Domains

The ER and SR scores capture two complementary aspects of macrostructural knowledge, where ER reflects the diversity or pluralism of patterns in a domain, while SR reflects the degree of similarity across groups. Together, they define four possible categories:

(i) **Low ER, High SR (LH)**: Few dominant patterns (low pluralism) that are widely shared across countries. Structurally, this resembles a fully connected graph or uniform clique. Domains such as house numbers fall here, as their distributions look similar across countries (Mukherjee et al., 2024).

(ii) **High ER, Low SR (HL)**: Many patterns (high pluralism) but little cross-country similarity. The network resembles disconnected or random graphs (Bollobás & Bollobás, 1998), where each country follows its own path. Currency is a typical example, as each nation has its own system with little overlap.

(iii) **High ER, High SR (HH)**: Many patterns, yet countries form affinity clusters. The network resembles small-world structures with a dense core and peripheral clusters (Newman, 2000). Food-related domains illustrate this case, where global staples coexist with strong local variations.

(iv) **Low ER, Low SR (LL)**: Few dominant patterns, but countries split into separate clusters rather than forming a universal consensus. The network fragments into distinct agreement clusters, each centered on a different core. Religion is a representative case, with a handful of world religions dominating, but countries grouping into different blocks.

To validate these categories, we conducted a human study across nine cultural domains: convenient food, healthy food, common food, national dish, house numbers, religion, holidays, language, and currency. Participants ranked domains by the expected uniqueness of item distributions across countries. Unlike microstructural methods that require fine-grained responses (e.g., preferred food items), our approach only needs rankings of domain-level variation (e.g., how common currencies are relative to foods or languages), making macrostructural data collection more robust and efficient.

## 3.2 Collecting Ground Truth Data

Let $R$ be a ranking $d_i \geq d_j ... \geq d_m$ over the set of domains $D$, ordered by how prevalent or consensus-driven participants expect the answers to domain questions $Q^{d_i}$ to be across countries $C$. To obtain a human-grounded ranking $R$, we conducted a survey with 80 participants from 16 world regions (5 per region; details in Section B, Appendix A). Each participant was asked to rank 9 domain-specific questions, each chosen to concisely capture the domain's observable property. After two rounds of pilot testing with 10 internal graduate-level participants, the final survey was hosted on Google Forms and administered via Prolific, and included two attention checks to ensure response quality (details in Section B). Participants were compensated at an hourly rate of £9 (well above Prolific's recommended pay guidelines) and took 5-10 minutes each to complete the survey.

Table 1 presents the 9 questions that participants ranked, and the corresponding ranking by averaging the scores across all participants. We observe that participants generally judged domains like convenient food and house numbers to be more common across countries than domains like national dish or currency, reflecting higher expected consensus (SR) in the former and greater divergence in the latter. To estimate expected pluralism (ER), three pilot participants annotated the likely diversity of items per domain and reached consensus. Table 1 (col. category) summarizes both ER-SR dimensions along with the number of items used for LLM evaluation in the following experiments.

| # | Domain | Question | Avg Score | Std Dev | Category | # Items |
|---|--------|----------|-----------|---------|----------|---------|
| 1 | House numbers | House numbers in your country | 3.18 | 2.23 | LH | 1000 |
| 2 | Convenient foods | Easy to prepare/buy foods in your country | 2.90 | 2.01 | HH | >3700 |
| 3 | Common foods | Commonly eaten foods in your country | 3.92 | 1.86 | HH | >3700 |
| 4 | Healthy foods | Healthy foods in your country | 3.98 | 1.62 | HH | >3700 |
| 5 | Religions | Major religions practiced in your country | 4.70 | 2.33 | LL | 21 |
| 6 | Holidays | Holidays and festivals celebrated in your country | 5.64 | 1.73 | LL | 2500 |
| 7 | Languages | Most common languages spoken in your country | 5.89 | 2.28 | LL | 161 |
| 8 | National dish | National dish of your country | 6.59 | 2.17 | HL | >3700 |
| 9 | Currency | Currency used in your country | 8.20 | 1.80 | HL | 168 |

Table 1: Domain Categorization by Humans

## 3.3 EXPERIMENT SETUP

For each cultural question, we curated an exhaustive list of candidate items spanning 170 countries from diverse sources, as detailed in Table 2. For every question $q_j \in Q^{d_i}$, we constructed a prompt, yielding $m \times k$ prompts across all domains. For each prompt, we varied the country and extracted the model's log probabilities over the full item list. This approach captures the model's uncertainty about unlikely country–item associations, which would be hidden if only final decoded responses were considered. When item names were split (by the tokenizer) into multiple tokens, we summed their log probabilities. The final country-level distribution for each question was obtained by applying softmax over all item log probabilities. These distributions were then stacked into matrices as defined in Section 3.1, and the country–country adjacency matrices of size $170 \times 170$ were computed using Equation 1. For each adjacency matrix, we calculated the ER and SR metrics (Equation 2). The ER and SR values were binarized as high or low relative to their medians across all questions and models, and the resulting labels were concatenated to assign each domain-question pair to one of the four ER-SR categories. Figure 1 illustrates our experiment pipeline in detail.

We experimented with 8 different-sized open-weight language models, detailed in Table 3. We experimented with $k = 3$ questions from each of the $m = 9$ domains, resulting in 27 prompts in total (details in Section B.2). The experiments were run using vLLM (Kwon et al., 2023) with FP16 quantization, on 2x NVIDIA RTX 6000 GPUs running for 50 GPU hours.

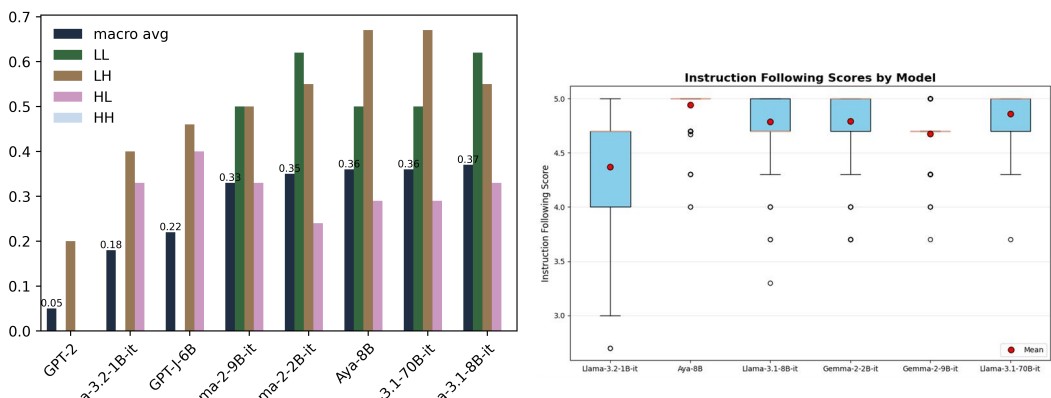

Figure 2: Model & Category-wise Macro-F1.

Figure 3: LLMs' instruction following capacity.

## 4 RESULTS

### 4.1 COMPARISON WITH HUMAN GROUND TRUTH DATA

For each model, we compare model-assigned ER-SR categories with human-derived ground truth data (Section 3.2) and report macro-F1 scores in Figure 2, along with the category-wise scores. We observe the following: (i) Llama-8B performs best, followed by Llama-70B and Aya-8B. Interestingly, Llama-70B does not clearly outperform its much smaller parameterized 8B counterpart,

suggesting that **macrostructural competence may not scale with model size** as it does on existing benchmarks like MMLU (Hendrycks et al., 2020), SQuAD (Rajpurkar et al., 2016), GSM-8K (Cobbe et al., 2021), etc. (ii) Similarly, Gemma-2-2B and 9B versions perform similarly, leading us to conjecture that **instruction-tuned models may plateau in macrostructural ability beyond a certain size**, which in our case is at 2B parameters. (iii) Older and smaller models, such as GPT-2, perform substantially worse than newer instruction-tuned LLMs and even larger non-instruct models like GPT-J-6B. (iv) Llama-3.2-1B performs above GPT-2 but below GPT-J-6B, supporting the idea that **a certain parameter threshold is necessary before macrostructural performance stabilizes**. (v) Strikingly, **all eight models consistently misclassify food domains as HL rather than HH**, treating national cuisines as disconnected despite humans expecting both diversity and cross-country similarity. This systematic failure across models suggests that current LLMs fail to internalize the relational structure of food culture. **They possibly learn discrete and stereotypical national categories** without regional clustering or shared culinary elements. (vi) Llama-3.2-1B and GPT-J-6B only predict HL and LH, missing nuanced HH/LL categories, whereas GPT-2 always defaults to LH, failing to even acknowledge cultural variations. In Figure 2, **we observe an increase in the number of correctly predicted categories as model complexity increases from fewer parameterized and less complex (left) to more parameterized and more complex (right)**. Although it eventually plateaus, where no model can correctly classify the HH category. We further illustrate a heatmap of the ER and SR values for all models and questions in Figures 5 and 6.

### 4.2 Model ranking based on Microstructures vs. Macrostructures

To compare insights from macrostructural versus microstructural analyses, we examined recent cultural benchmarks published in the last three years and extracted model rankings by domain. We found five benchmarks that overlap with some of our nine domains and eight models. Table 5 lists the benchmarks and the ranking among models from each paper. We observe that Llama-3.1-8B-it is consistently ranked higher than Aya-8B in existing studies, which aligns with our results. However, unlike our results, these studies rank Gemma-2 variants above both Llama and Aya. They also report Llama-70B as performing significantly better than others, whereas our macrostructural analysis suggests that performance asymptotes in instruction-tuned models beyond 2B parameters (Gemma-2-2B-it). These discrepancies highlight how macrostructural evaluation reveals different aspects of model knowledge and cultural representation than microstructural methods, which focus on point-wise factual accuracy.

## 5 META-CULTURE IN PRACTICE: A SIMULATION-BASED EXPERIMENT

We designed a simulation study to test whether macrostructure-based performance is a useful indicator of meta-cultural competency in downstream tasks. As a case study, we focused on recipe recommendation, a common application of AI systems (Yang et al., 2024; Lyu et al., 2023), where cultural knowledge, personalization, and explication play central roles.

### 5.1 Setup

**User Simulation:** To scale the setup, we replaced human users with an LLM-based simulator. We first collected breakfast-habit data from 10 participants (the same pilot participants in Section 3.2), representing 5 distinct cuisines and a range of dietary profiles: vegetarian, non-vegetarian, halal, diabetic, varying cooking skills, and nutrition goals. Each participant interacted with GPT-5 (ChatGPT) through a structured prompt (in Box 1 and Box 2) that instructed GPT-5 to act as an anthropologist and elicit detailed breakfast-related habits. The participants reviewed and edited the resulting JSON persona summaries to ensure accuracy and authenticity. These finalized personas formed the basis for the GPT-4o user simulator (prompt in Box 3).

**Role-Play Setup:** In the experiment, the user simulator (GPT-4o) interacted with the six instruction-tuned LLMs from Section 3.3, acting as recipe recommenders (using prompt in Box 5). The recommender was free to ask clarifying questions (as mentioned in the prompt), and the conversation ended once a breakfast recipe was recommended. Both the simulator and recommenders were run with temperature 1, and each scenario was repeated 10 times, to yield conversational variations.

**Evaluation:** Conversations were rated by GPT-4o as a judge (using prompts in Box 6 and Box 7) on two dimensions, each from 1 (low) to 10 (high), reflecting meta-cultural competency: (i) **Appropriateness (APR):** The alignment of the final recommendation with the user persona (preferences, cultural background, dietary restrictions, lifestyle, etc). High scores reflect high meta-cultural competency (variational awareness and strong explication strategy) and cultural awareness. (ii) **Interaction (INT):** The quality of clarifying questions used to elicit user-specific needs, such that it could personalize the recommendation. High scores reflect high explication (operationalizing variational awareness) and instruction following capacity.

**Control Setup:** To isolate a model's default capacity of operationalizing its variational awareness via explication, we also ran a control condition where recommenders were not explicitly nudged to ask clarifying questions. We omitted the instruction from the prompt that specified how food preferences are individual-specific and dependent on several personal factors, as illustrated in the prompt in Box 4. Each model engaged in 200 conversations (100 control, 100 experiment), totaling 1200 conversations across six models.

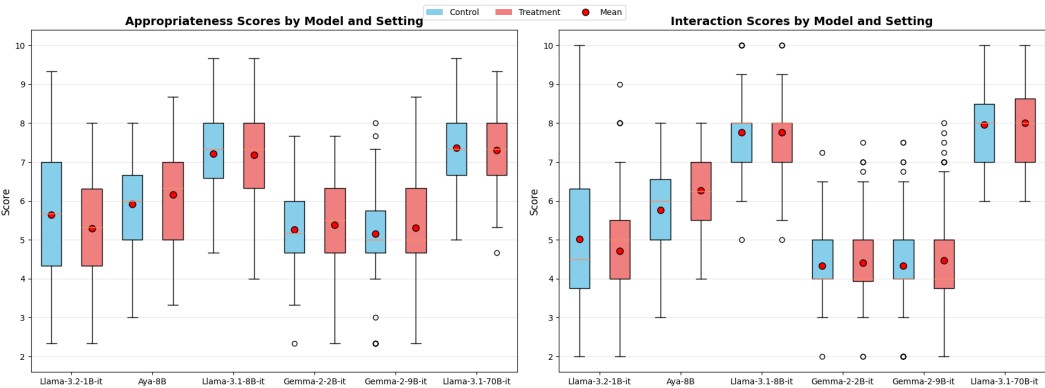

Figure 4: Model-wise APR (**L**) and INT (**R**) scores for control and treatment groups.

## 5.2 RESULTS

Figure 4 presents the results of the simulation study, with models on the x-axis in ascending order by their microstructural cultural awareness (Section 4.2). We highlight four main observations: (i) **Llama-3.1-70B-it and Llama-3.1-8B-it perform best.** Both models achieve the highest APR and INT scores. Their performance in the control and treatment groups shows no statistically significant difference, suggesting that they are intrinsically variationally aware and meta-culturally competent to an extent, and already possess strong explication skills without requiring explicit instructions. (ii) **Gemma-2 models perform below Llama variants** ($p < 0.05$), despite showing strong cultural awareness in prior microstructural benchmarks. This drop aligns with their weaker macrostructural rankings (Section 4.1), underscoring the strength of the APR metric in capturing meta-cultural competency holistically. Models with low variational awareness cannot effectively explicate and thus fail to achieve high APR scores. Similar to Llama, Gemma shows no significant difference between control and treatment, though Gemma-2B exhibits a small, non-significant APR improvement. (iii) **Aya-8B benefits from nudging.** Unlike Llama and Gemma, Aya shows a statistically significant improvement in the treatment condition, indicating that it does not display meta-cultural competency by default but can be nudged to do so. Interestingly, Aya also outperforms Gemma variants, which is consistent with its macrostructure-based ranking but contrary to its microstructural scores from other benchmarks. (iv) **Llama-3.2-1B-it shows mixed behavior.** Its performance lies between Aya and Gemma and is much lower than other Llama variants. Furthermore, its performance does not change with adding more instructions in the treatment prompt, suggesting a possible lack of following instructions and limited meta-cultural competency.

**Instruction Following vs. Explication.** To test whether INT reflects genuine explication capacity rather than simple instruction following, we used GPT-4o as a judge (using Prompts Box 8 and Box 9). Figure 3 shows that all models scored well above 4.5/5 on instruction following, except

Llama-3.2-1B-it (4.4), which significantly ($p < 0.05$) lags behind others, while demonstrating a high variability. This indicates that differences in INT are not explained by instruction adherence alone, but by models' ability to operationalize their variational awareness and explicate appropriately. Otherwise, we would have seen all models attain similar INT scores.

**Human Validation.** We further validated the results with human evaluations. From the 1,200 simulated conversations, we randomly sampled 70 conversations spanning all models and both conditions. Seven of the 10 participants who originally provided persona data rated their respective simulated conversations on a 1-5 scale for: (i) **Persona alignment**: How well GPT-4o represented their persona? (ii) **Interaction (INT) alignment**: How well did the recommender gather relevant information before recommending? (iii) **Appropriateness (APR) alignment**: How well did the final recipe match the user's preferences, as depicted through the persona? To avoid bias, evaluators were blinded to the source of each conversation. Detailed instructions are provided in Appendix Box 10. GPT-4o achieved an average persona alignment score of 4.3/5, **indicating adequate representation of user personas**. For human validation of INT and APR scores, we compare the ratings of all possible pairs of conversations by GPT-4o and the human evaluator. If the ratings order the conversations in the opposite direction (e.g., conversation A higher than conversation B for GPT-4o and the reverse for human), then we consider it a misalignment. We observe that 22.8% and 16.7% of the pairs were misaligned (corresponding to a Cohen's Kappa of 0.54 and 0.72) for APR and INT, respectively. We also compute the Spearman's rank coefficient between the model rankings as obtained from the average scores given by human evaluators and GPT-4o, and find the values as 0.83 and 0.89 for APR and INT, respectively. These results indicate that, while individual conversation ratings are sometimes noisy due to the inherent subjectivity of the task, GPT-4o reliably captures the relative ranking of the models' performances at an aggregate level.

## 6 DISCUSSION AND CONCLUSION

Our human evaluation shows that while judgments aligned with GPT-4o as a judge, participants were underwhelmed by the recommendations. They reported recommendations were often repetitive or shallow, such as offering only minor variations of a single dish. This reflects a tendency to collapse diverse cultural and individual needs into stereotyped variants, a limitation noted in prior critiques of LLM cultural reasoning (Shen et al., 2024; Khan et al., 2025). This limitation is not surprising since our results (in Section 4.1) show that instruction-tuned LLMs plateau in macrostructural ability beyond roughly 2B parameters, with even 70B models failing across all classes, indicating that scale alone is insufficient. This raises deeper questions about whether current training regimes, which are dominated by large-scale pretraining and instruction tuning, can ever yield models with robust meta-cultural competency. As Saha et al. (2025) argues, achieving genuine cultural sensitivity in AI may require a paradigm shift in training and evaluation, where models are explicitly optimized for relational, cross-cultural, and pluralistic knowledge structures rather than only factual correctness.

Our framework, while novel, has several constraints. First, we restricted our analysis to nine cultural domains. While these domains were selected to capture a range of structural topologies, they do not exhaust the breadth of cultural practices and knowledge structures. Second, our experiments were limited to open-weight models up to 70B parameters and a single downstream task of recipe recommendation. It remains an open question whether larger closed-source models, or different downstream use cases, would follow the same trends or break the observed asymptotes in macrostructural ability (Section 4.1). Third, our experiments are limited to only one language, that is, English, and therefore do not necessarily reflect the macrostructural properties of the models when probed in other languages, even though ideally the macrostuctures should be independent of the language.

Despite these limitations, our study demonstrates that macrostructural evaluation offers a powerful lens to examine variational awareness and meta-cultural competency in models. Also, a key advantage of our approach is that to estimate macrostructure, we do not need ground-truth references for specific cultures. However, since the macrostructure emerges only from a correct understanding of each individual culture and their interaction, an accurate or closer to the real-world macrostructure provides indirect evidence for accurate microstructural knowledge. The reverse is also true to some extent, but that would require benchmarking on a large number of cultural datasets from various groups. Nonetheless, micro and macrostructures, pertaining to distinct kinds of information, should be viewed as complementary means to understand models' cultural awareness.

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

# A  APPENDIX

## A.1  DATA COLLECTION

| Domain | Source |
|---|---|
| House Numbers | 1-1000 |
| Convenient foods
Common foods
Healthy foods
National dish | https://www.tasteatlas.com/, Worldcuisines (Winata et al., 2024) |
| Holidays | https://www.timeanddate.com/holidays/ |
| Languages | https://www.worldvaluessurvey.org/WVSDocumentationWV7.jsp |
| Religions | https://en.wikipedia.org/wiki/Major_religious_groups |
| Currencies | https://en.wikipedia.org/wiki/List_of_circulating_currencies |

Table 2: Domains and their sources of items.

The items for most of the domains were scraped from online sources using the BeautifulSoup[2] Python library. Table 2 lists the sources for each domain. The food-related questions were sourced from the *Food Choice Questionnaire* (FCQ) (Steptoe et al., 1995), which is designed to capture the diverse food-related behaviors across countries. The house number questions were inspired by Mukherjee et al. (2024). The remaining questions were inspired by the World Values Survey Questionnaire[3] and existing cultural evaluation benchmarks mentioned in Section 2.2.

# B  COLLECTING HUMAN DATA

We collected data from 80 participants from the following 16 geographic regions (5 per region): Arabic (encompassing Algeria, Bahrain, Iraq, Jordan, Kuwait, Lebanon, Libya, Mauritania, Morocco, Oman, Palestinian Territory, Qatar, Saudi Arabia, Somalia, Sudan, Syrian Arab Republic, Tunisia, Yemen), Australia, Bantu (Angola, Malawi, Mozambique, Tanzania, Zambia, Zimbabwe), Brazil, China, France, India, Indonesia, Japan, Mexico, Niger-Congo (Benin, Burkina Faso, Cameroon, Cote d'Ivoire, Gambia, Ghana, Guinea, Liberia, Nigeria, Senegal, Sierra Leone, Togo), Russia, Sweden, Turkic (Kazakhstan, Kyrgyzstan, Turkey, Uzbekistan), UK, and the USA. We used Prolific[4] to disseminate the survey and used Google Forms to collect the responses. The survey detailed the task and provided clear annotation guidelines along with examples, as depicted in Figure 7. The main survey had only one question, illustrated in Figure 8, which asked participants to rank the nine cultural domains based on how common they expect the elements of the domain to be across countries and cultures. Finally, the survey had three attention check questions, illustrated in Figure 9, which captured whether the participants understood the guidelines properly and whether the rankings were valid. Participants failing the attention checks were discarded.

| Model | HuggingFace ID | Parameters | Instruct-Tuned |
|---|---|---|---|
| Aya-8B | CohereLabAI/aya-23-8B | 8B | ✓ |
| Gemma-2-2B-it | google/gemma-2-2b | 2B | ✓ |
| Gemma-2-9B-it | google/gemma-2-9b | 9B | ✓ |
| GPT-2 | openai-community/gpt2 | 124M | × |
| GPT-J-6B | EleutherAI/gpt-j-6b | 6B | × |
| Llama-3.1-70B-it | meta-llama/Meta-Llama-3.1-70B | 70B | ✓ |
| Llama-3.1-8B-it | meta-llama/Meta-Llama-3.1-8B | 8B | ✓ |
| Llama-3.2-1B-it | meta-llama/Llama-3.2-1B | 1B | ✓ |

Table 3: Overview of evaluated language models and their characteristics.

---

[2]https://pypi.org/project/beautifulsoup4/

[3]https://www.worldvaluessurvey.org/WVSContents.jsp

[4]https://www.prolific.com/

## B.1 ADDITIONAL ANALYSIS

| Model | Correlation |
|---|---|
| Llama-3.1-70B-it | -0.64 |
| Gemma-2-2B-it | -0.49 |
| GPT-J-6B | -0.48 |
| Llama-3.2-1B-it | -0.45 |
| Gemma-2-9B-it | -0.42 |
| GPT-2 | -0.39 |
| Llama-3.1-8B-it | -0.29 |
| Aya-8B | -0.28 |

Table 4: Model-wise correlation between ER and SR across all questions

We compute the correlation between the ER and SR scores for each model, which ideally should be negatively correlated. We observe (in Table 4) that Llama-3.1-70B-it mimics this pattern the best. Interestingly, the 8B versions of Llama-3.1 and Aya are least negatively correlated, even below GPT-2 and GPT-J. Nonetheless, all models exhibit a negative correlation, indicating a basic level of macrostructural knowledge. Figures 5 and 6 illustrate a heatmap of the ER and SR values for all models and questions.

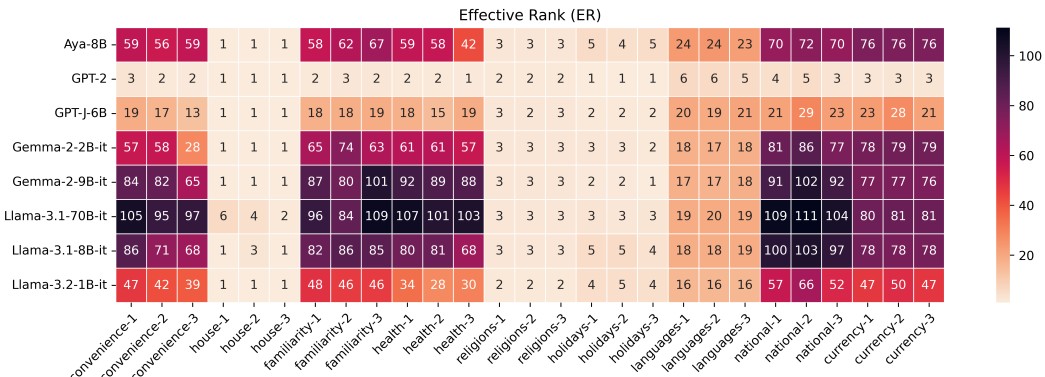

Figure 5: Heatmap of the Effective Rank (ER) across all models and questions. Lower = Low plurality; Higher = High plurality.

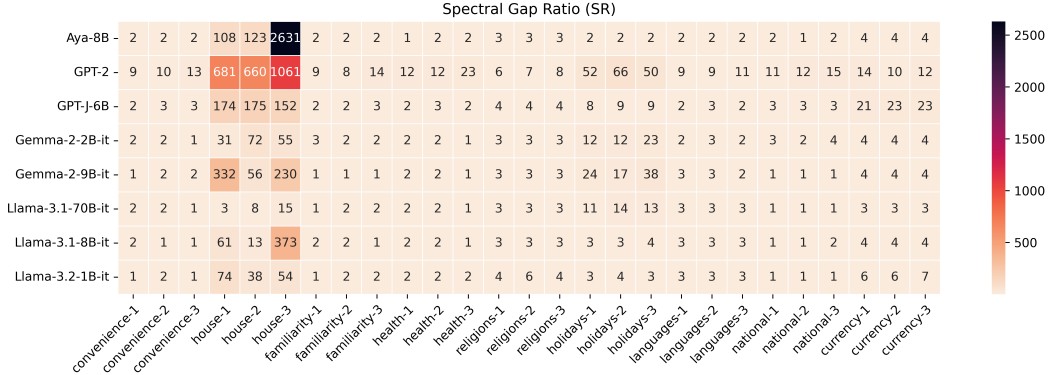

Figure 6: Heatmap of the Spectral Ratio (SR) across all models and questions. Lower = Low consensus; Higher = High consensus.

---

[1]Benchmark results for BLEnD and CAMeL-2 obtained from (Yu et al., 2025).

| Benchmark | Domain | Ranking |
|---|---|---|
| CultureBench (QA) (Chiu et al., 2025) | 17 topics under 3 categories: Daily life (Food, Language/Communication, Family, Clothing, etc.), Social Etiquette, Wider Society (Celebrations, Religion, Politics) | Aya 8B < LLaMA 3.1 8B < Gemma 2 9B < LLaMA-3.1-70B |
| BLEnD (QA)[1] (Myung et al., 2024) | 6 categories: food, sports, family, education, holidays, work-life | Aya-8B < LLaMA-3.1-8B |
| CAMeL-2 (Extractive QA)[1] (Naous & Xu, 2025) | Location, Beverage, Food, Sports | Aya-8B < LLaMA-3.1-8B |
| CULEMO (Emotion prediction) (Belay et al., 2025) | Family relationships, Social etiquette, Cultural/religious practices, Emotional/psychological situations | Aya-8B < LLaMA-3.1-8B < Gemma 2 2B *(no clear winner between Gemma models)* |
| IndoCulture (QA) (Koto et al., 2024a) | 12 cultural topics: Food, Wedding, Art, Family Activities, etc. | *(Relatively older models)* mT0 < BLOOMZ < LLaMA-3-8B < LLaMA-3-70B *(LLaMA-3.1 not specified)* |

Table 5: Model Rankings across Cultural Understanding Benchmarks (Microstructures)

## B.2 MACROSTRUCTURAL ANALYSIS EXPERIMENTAL SETUP

Our macrostructural evaluation framework requires systematic prompt construction to elicit consistent responses across cultural domains and countries. We employ domain-specific prompt templates that combine a prefix instruction with a main query template.

For instruction-tuned models, we construct the user message by concatenating the prefix prompt and main prompt: "{prefix_prompt} {main_prompt}". This combined prompt is then processed through each model's corresponding chat template to ensure proper formatting. For base models without instruction tuning (GPT-2 and GPT-J-6B), we append an explicit completion cue: "{prefix_prompt} {main_prompt}\nAnswer:". This additional prompt engineering helps guide these models toward generating the desired response format.

The prefix prompts provide consistent instructions for response formatting across all domains, while the main prompts contain the specific cultural queries with country placeholders. Tables 6 and 7 present the complete set of prefix and main prompt templates used to elicit country-specific responses across our nine cultural domains.

| Domain | Prefix Prompt |
|--------|---------------|
| Convenient/ Common/ Healthy foods | Answer the given question related to food items in a country. Generate only the name of the food items. Do not output any extra text. |
| House numbers | Answer the given question related to house numbers in a country. Generate only the house number. Do not output any extra text. |
| Languages | Answer the given question related to languages in a country. Generate only the name of the languages or dialects. Do not output any extra text. |
| National dish | Answer the given question related to food items in a country. Generate only the name of the food item. Do not output any extra text. |
| Religions | Answer the given question related to religions in a country. Generate only the name of the religions. Do not output any extra text. |
| Currency | Answer the given question related to currency in a country. Generate only the name of the currency. Do not output any extra text. |
| Holidays | Answer the given question related to holidays in a country. Generate only the name of the holidays. Do not output any extra text. |

Table 6: Prompt prefix templates used across cultural domains.

---

**Box 1: Prompt for Human Breakfast persona generation using GPT-5**

```
You are a friendly interviewer who is helping to build a detailed persona about my
    breakfast habits.

Instructions:
1. Ask me questions in a natural, conversational way about:
   - The breakfast foods I usually eat.
   - How I prepare or cook them.
   - Why I prefer those choices (taste, convenience, health, tradition, family habits, etc
     .).
   - Variations across weekdays vs weekends.
   - Any cultural, personal, or childhood influences on my breakfast choices.
   - My feelings or associations with breakfast (comfort, energy, social, etc.).
   - Health considerations, including conditions, allergies, or dietary restrictions.
   - My current lifestyle and routines that affect breakfast.
   - Ask me any other details that might help create a comprehensive persona.
2. Follow up naturally on my answers. Don't just move to the next question dig deeper if
     something is interesting or unique.
3. Keep the tone casual, like two people chatting. It should feel like a real conversation,
      not a survey.
4. I will end the conversation by typing: "That's all for now".
   - When I do this, please stop asking questions.
   - Then, **summarize the conversation** and create a **Python dictionary** representing
      my breakfast persona.

5. The dictionary must be a valid Python dictionary and structured as follows:
{Please refer to Prompt Box 2 for the persona dictionary format}

1. Output the dictionary only, nothing else, after my final message.
2. Always ask questions in the conversation so that you can fill in all fields as
    completely as possible.
```

| Domain | Var. | Prompt Template |
|---|---|---|
| Convenient foods | 1 | The food in {country} that is easy to prepare is |
| | 2 | The food in {country} that takes no time to prepare is |
| | 3 | The food in {country} that can be bought in shops close to where someone lives or works is |
| Common foods | 1 | The food in {country} that people usually eat is |
| | 2 | The food in {country} that is familiar is |
| | 3 | The food in {country} that is like the food someone ate when they were a child is |
| Healthy foods | 1 | The food in {country} that keeps someone healthy is |
| | 2 | The nutritious food in {country} is |
| | 3 | The food in {country} that is good for someone's skin/teeth/hair/nails etc is |
| House numbers | 1 | The typical house numbers in {country} are |
| | 2 | The most common house numbers in {country} are |
| | 3 | In {country}, the house numbers that are frequently used are |
| National dish | 1 | The national food of {country} is |
| | 2 | The national dish of {country} is |
| | 3 | In {country}, the national food is |
| Religions | 1 | The major religions practiced in {country} are |
| | 2 | The prominent religions in {country} include |
| | 3 | In {country}, the most common religions are |
| Currency | 1 | The currency used in {country} is |
| | 2 | The official currency of {country} is |
| | 3 | In {country}, the currency in circulation is |
| Languages | 1 | The major languages spoken in {country} are |
| | 2 | The prominent languages in {country} include |
| | 3 | In {country}, the most common spoken languages are |
| Holidays | 1 | The holidays celebrated in {country} include |
| | 2 | In {country}, the celebrated holidays are |
| | 3 | The holidays observed in {country} include |

Table 7: Prompt templates used across cultural domains for eliciting country-specific responses.

**Box 2: Human Breakfast Persona Format**

```
{
    "name": "Generated by ChatGPT or left blank",
    "demographics": {
        "age": None,
        "gender": None,
        "location": None,
        "occupation": None,
        "household": "Lives alone / with family / roommates etc.",
        "lifestyle": "sedentary / active / mixed",
        "health_conditions": []
    },
    "typical_breakfast": ["list of foods usually eaten"],
    "preparation_style": ["list or description of how items are prepared"],
    "preferences": {
        "reasons": ["health", "convenience", "taste", "tradition"],
        "weekday_vs_weekend": "differences if any",
        "preferred_beverages": ["tea", "coffee", "juice"],
        "favorite_items": ["specific breakfast dishes they like"],
        "disliked_items": ["foods they avoid at breakfast"]
    },
    "constraints": {
        "time_available": "short / moderate / long",
        "dietary_restrictions": ["vegetarian", "allergies"],
        "budget": "low / medium / high",
        "availability_of_items": "easy / seasonal / hard-to-find",
        "cooking_skills": "novice / intermediate / expert"
    },
    "cultural_influences": {
        "childhood_habits": ["breakfasts they grew up with"],
        "family_traditions": ["family-specific practices"],
        "regional_influences": ["foods common to their culture"],
        "religious_or_festive_influences": ["festival breakfasts"]
    },
    "emotional_associations": ["comfort", "energy", "routine"],
    "routines_and_context": {
        "timing": "early morning / mid-morning / varies",
        "who_with": "alone / with family / socially",
        "where": "home / office / cafe",
        "pace": "leisurely / rushed / on-the-go"
    },
    "aspirations_or_changes": {
        "desired_changes": "healthier / more variety / faster",
        "ideal_breakfast": "description of dream breakfast"
    },
    "health_and_wellbeing": {
        "perceived_healthiness": "healthy / balanced / indulgent",
        "impact_on_day": "effects on mood, energy, productivity",
        "skipping_habits": "never / sometimes / often skip"
    },
    "social_and_personality_signals": {
        "identity_connection": "cultural reflection",
        "social_sharing": "eating with others, posting online",
        "personality_traits_visible": "organized, spontaneous"
    },
    "notable_quotes": ["memorable things said during chat"],
    "summary": "Short paragraph capturing breakfast persona"
}
```

```
Box 3: Persona Simulation Propmt
```

```
You are an AI assistant skilled at role-playing a user persona.

You will be provided a user persona collected from real users, which includes details about
    their breakfast habits, preferences and other cultural details. Your task is to
    conversate with a breakfast recipe recommender, simulating the persona as
    authentically as possible.
This is a role-playing exercise, where the end goal is to judge how well the recommender
    tries to understand and adapt to your persona (cultural personality) to finally
    recommend breakfast recipes that suit your tastes and needs.

Rules when interacting with the recipe generator:

- Include influences such as health, culture, routine, childhood memories, or lifestyle
    where relevant.
- Express your feelings toward breakfast (comfort, energy, social aspects, etc.).
- Stay consistent with your persona at all times.
- Do not divulge extra information if not asked for. Let the recommender ask for more
    details, do not overshare. Only share details a real person would naturally share.
- Do not share names of dishes you eat. Let the recommender recommend dishes.
- Your response should not be more than 100 words.

Your Persona Details:
```
{persona}
```
```

```
Box 4: Breakfast Recommender Prompt Control
```

```
You are an AI assistant that specializes in creating personalized breakfast recipes.

Your end-goal is to recommend a recipe that the user can realistically use in their daily
    life. To do so, you can converse with the user to understand their breakfast
    preferences and habits.

When you recommend a recipe, provide:
- #Dish name: {{The name of the dish}}
- #Ingredients: {{Ingredients (with approximate quantities) in 50 words}}
- #Instructions: {{Step-by-step preparation instructions (simple and practical) in 50 words
    }}

Rules when interacting with the user:
- Be polite, conversational, and efficient.
- Note that the conversation ends once you provide a recipe. You cannot iterate after that.

Start the conversation with the user.
```

```
Box 5: Breakfast Recommender Prompt Treatment
```

```
You are an AI assistant that specializes in creating personalized breakfast recipes.

Your end-goal is to recommend a recipe that the user can realistically use in their daily
    life. To do so, you can converse with the user to understand their breakfast
    preferences and habits.

When you recommend a recipe, provide:
- #Dish name: {{The name of the dish}}
- #Ingredients: {{Ingredients (with approximate quantities) in 50 words}}
- #Instructions: {{Step-by-step preparation instructions (simple and practical) in 50 words
    }}

Rules when interacting with the user:
- Be polite, conversational, and efficient.
- Note that the conversation ends once you provide a recipe. You cannot iterate after that.
- Remember that food preferences are highly individual-specific, which depends on a variety
    of features such as health conditions, lifestyle, culture, routine, location, cooking
    skill level, preferences, etc.

Start the conversation with the user.
```

Box 6: LLM as Judge for Appropriateness and Interaction quality prompt (Part A)

```
AI Rules
- Output response in a valid JSON format.
- Do not wrap the JSON codes in JSON or Python markers.
- JSON keys and values in double-quotes.

# Description
You are an impartial evaluator tasked with judging how well a recommended breakfast recipe
    fits a given user persona and demonstrates meta-cultural competence.

**META-CULTURAL COMPETENCE** = The ability to understand that preferences are deeply
    personal, shaped by culture, constraints, and individual circumstances, and to adapt
    recommendations accordingly.

You will be given:
1. Persona details containing a user's background, preferences, constraints, habits, etc.
2. Conversation transcript between the user and the recipe recommender. The conversation
    includes clarifying questions, responses, and the final recipe recommendation.

# Task
## CULTURAL DEPTH ANALYSIS
Examine: Did the recommender understand the user's cultural background and incorporate it
    meaningfully?

**Cultural cues present in persona:** [Identify specific cultural elements from persona]
**Cultural cues mentioned by user:** [What cultural information did user share in
    conversation?]
**Recommender's cultural probing:** [What questions did they ask about background/
    traditions?]
**Cultural incorporation:** [How did final recipe reflect cultural understanding?]

SCORING CRITERIA:
- EXCELLENT (8-10): Understood cultural background, family traditions, regional influences;
     incorporated cultural elements naturally; showed awareness that food habits vary by
    culture/region; adapted language and suggestions to cultural context
- POOR (1-4): Did not understand cultural background; gave generic Western recommendations
    regardless of user's origin; ignored cultural cues provided by user; assumed universal
     food preferences

**Preliminary Cultural Depth Score:** ___

## CONSTRAINT AWARENESS ANALYSIS
Examine: Did the recommender understand and respect the user's practical limitations?

**Key constraints from persona:** [List time, skill, equipment, lifestyle constraints]
**Constraints mentioned by user:** [What limitations did user express?]
**Recommender's constraint probing:** [What questions about practical limitations?]
**Constraint accommodation:** [How did recipe match user's capabilities?]

SCORING CRITERIA:
- EXCELLENT (8-10): Asked about time constraints, cooking skills, available ingredients;
    tailored complexity to match user's capabilities; considered lifestyle factors;
    balanced ideals with practical realities
- POOR (1-4): Ignored stated time/skill constraints; suggested elaborate recipes for rushed
     mornings; failed to ask about practical limitations; gave impractical recommendations

**Preliminary Constraint Awareness Score:** ___

## PERSONALIZATION QUALITY ANALYSIS
Examine: How well did the final recommendation match the user's specific preferences and
    needs?

**User's stated preferences:** [List favorite foods, preparation styles, etc.]
**User's dislikes/restrictions:** [What to avoid]
**User's goals/aspirations:** [Health goals, ideal breakfast, etc.]
**Recipe alignment:** [How well does final recipe match these elements?]

SCORING CRITERIA:
- EXCELLENT (8-10): Recipe aligns with stated favorite foods/preparation styles; respects
    dietary preferences and health goals; matches preferred meal timing and social context
    ; incorporates emotional associations
- POOR (1-4): Generic recommendation ignoring stated preferences; contradicts user's
    typical habits or goals; misses obvious preference signals; one-size-fits-all approach
```

```
Box 7: LLM as Judge for Appropriateness and Interaction quality prompt (Part B)

**Preliminary Personalization Score:** ___

## INTERACTION PROCESS ANALYSIS
Examine: Did the recommender gather sufficient information through quality questioning?

**Number of questions asked:** ___
**Types of questions:** [Cultural? Constraints? Preferences? Follow-ups?]
**Question quality:** [Focused vs generic? Built on responses?]
**Information gathering progression:** [Did understanding develop over turns?]

SCORING CRITERIA:
- EXCELLENT (8-10): Asked 3+ focused, relevant questions before recommending; built
    understanding progressively; asked follow-up questions based on responses; avoided
    generic/leading questions
- POOR (1-4): Jumped to recommendation without adequate questioning; asked irrelevant/
    superficial questions; failed to build on responses; used leading questions

**Preliminary Interaction Score:** ___

## PENALTY ASSESSMENT
Check for critical failures:
- [ ] Asked user to name specific dishes instead of understanding needs
- [ ] Completely ignored obvious cultural cues provided by user
- [ ] Recommended something directly contradicting stated constraints

**Penalties to apply:** [List any penalties and which scores they affect]

# STEP 2: FINAL SCORING

Based on the analysis above, provide final scores and reasoning:

**FINAL CULTURAL DEPTH SCORE:** ___
**Reasoning:** [Synthesize analysis into final justification]

**FINAL CONSTRAINT AWARENESS SCORE:** ___
**Reasoning:** [Synthesize analysis into final justification]

**FINAL PERSONALIZATION QUALITY SCORE:** ___
**Reasoning:** [Synthesize analysis into final justification]

**FINAL INTERACTION PROCESS SCORE:** ___
**Reasoning:** [Synthesize analysis into final justification]

**APPROPRIATENESS SCORE** = Average of (Cultural Depth + Constraint Awareness +
    Personalization Quality) = ___

**OVERALL ASSESSMENT:** [2-3 sentence summary of recommender's meta-cultural competence]

# STEP 3: JSON OUTPUT

Now provide the structured JSON output:

{{
  "cultural_depth": {{"thought_and_reasoning": "...", "score": X}},
  "constraint_awareness": {{"thought_and_reasoning": "...", "score": X}},
  "personalization_quality": {{"thought_and_reasoning": "...", "score": X}},
  "interaction_process": {{"thought_and_reasoning": "...", "score": X}},
  "penalties_applied": ["penalty1", "penalty2"],
  "appropriateness_score": X.X,
  "interaction_quality_score": X,
  "overall_assessment": "..."
}}

---

User Persona Details:
```
{persona}
```

Conversation to evaluate:
```
{conversation}
```
```

---

**Box 8: LLM as Judge for Instruction Following Ability prompt (Part A)**

```
AI Rules
- Output response in a valid JSON format.
- Do not wrap the JSON codes in JSON or Python markers.
- JSON keys and values in double-quotes.

# Description
You are evaluating how well an AI model followed specific instructions in a breakfast
    recommendation conversation.

# EVALUATION CRITERIA

Rate each aspect (1-5 scale):

## 1. FORMAT COMPLIANCE (1-5)
**Check**: Did they provide the recipe in the exact requested format?

Required format:
- #Dish name: [name]
- #Ingredients: [~50 words with quantities]
- #Instructions: [~50 words, simple and practical]

**Scale**:
- **5**: Perfect format adherence, all sections present and correctly labeled
- **4**: Minor format issues (slightly over/under word count)
- **3**: Format mostly correct but missing labels or significant word count issues
- **2**: Major format problems, some sections missing or incorrectly structured
- **1**: No adherence to requested format

## 2. BEHAVIORAL RULES & CONVERSATION DYNAMICS (1-5)
**Check**: Did they follow interaction rules and maintain good conversational flow?

Required behaviors:
- Be polite, conversational, and efficient
- End conversation after providing recipe (no iteration)
- Start conversation appropriately
- Maintain natural conversational progression

**Scale**:
- **5**: Perfect conversational flow - polite, natural progression, followed all rules
- **4**: Good conversation with minor issues (slightly verbose or awkward transition)
- **3**: Adequate conversation but some mechanical issues
- **2**: Poor conversational mechanics (robotic, rude, or tried to continue after recipe)
- **1**: Very poor conversation (completely inappropriate or disregarded basic rules)

## 3. TASK FOCUS (1-5)
**Check**: Did they stay focused on the core task?

Required focus:
- Specialize in breakfast recipes
- Aim for realistic daily-life usage
- Understand user preferences/habits before recommending

**Scale**:
- **5**: Perfect task focus, stayed on breakfast recommendations throughout
- **4**: Good focus with minor diversions
- **3**: Adequate focus but some tangential content
- **2**: Significant diversions from breakfast recipe task
- **1**: Lost track of core task entirely

# ANALYSIS & REASONING

## Format Compliance Analysis:
[Analyze: Are all required sections present? Word counts appropriate? Labels correct?]

## Behavioral Rules Analysis:
[Analyze: Was tone appropriate? Did they end after recipe? Natural flow?]

## Task Focus Analysis:
[Analyze: Did they stay on breakfast recipes? Maintain daily-life practicality focus?]
```

---

**Box 9: LLM as Judge for Instruction Following Ability prompt (Part B)**

```
# OBJECTIVE CHECKS

## Format Check:
- Dish name present: Yes/No
- Ingredients section: Yes/No (Word count: ˜___)
- Instructions section: Yes/No (Word count: ˜___)
- Correct labels used: Yes/No

## Behavioral & Conversation Check:
- Conversational tone: Yes/No
- Ended after recipe: Yes/No
- Started appropriately: Yes/No
- Natural progression: Yes/No

## Word Count Analysis:
- Ingredients word count: ___
- Instructions word count: ___
- Target was ˜50 words each

Overal Instruction Score: Scale 1-5

# JSON OUTPUT

{{
  "format_analysis": "Detailed reasoning about format adherence and specific issues found",
  "behavioral_analysis": "Detailed reasoning about conversational behavior and rule
      following",
  "task_focus_analysis": "Detailed reasoning about staying on task and maintaining focus",
  "objective_checks": {{
    "dish_name_present": true/false,
    "ingredients_section_present": true/false,
    "instructions_section_present": true/false,
    "correct_labels_used": true/false,
    "appropriate_word_counts": true/false,
    "conversational_tone": true/false,
    "ended_after_recipe": true/false,
    "started_appropriately": true/false,
    "natural_progression": true/false,
    "stayed_on_breakfast_task": true/false
  }},
  "format_compliance": {{"score": X, "key_issues": ["issue1", "issue2"]}},
  "behavioral_rules": {{"score": X, "key_issues": ["issue1", "issue2"]}},
  "task_focus": {{"score": X, "key_issues": ["issue1", "issue2"]}},

  "overall_instruction_following": X.X,
  "summary": "Brief assessment of overall instruction following quality"
}}

---

Conversation to evaluate:
```
{conversation}
```
```

1458
1459
1460
1461
1462
1463
1464
1465
1466
1467
1468
1469
1470
1471
1472
1473
1474
1475
1476
1477
1478
1479
1480
1481
1482
1483
1484
1485
1486
1487
1488
1489
1490
1491
1492
1493
1494
1495
1496
1497
1498
1499
1500
1501
1502
1503
1504
1505
1506
1507
1508
1509
1510
1511

---

**Box 10: Human Validation for LLM-as-Judge Instructions**

```
TASK OVERVIEW
You will evaluate AI conversations about breakfast recommendations. Each task has two parts
    :
        1. Persona Alignment: How well did the "user" represent the given persona?
        2. Recommender Quality: How well did the "recommender" adapt to the user?

INSTRUCTIONS
STEP 1: READ THE PERSONA AND CONVERSATION
First, carefully read the persona details below. This represents your breakfast preferences
     and habits captured in the previous data collection exercise. You are also provided
    the conversation transcript between the user and the recipe recommender. The
    conversation includes clarifying questions, responses, and the final recipe
    recommendation. Read and refer it for all the below assessments.

STEP 2: EVALUATE PERSONA ALIGNMENT (1-5)
Question: How authentically did the AI user represent this persona in the conversation?
Look for:
        Did they mention relevant cultural background when appropriate?
        Did they express the right constraints (time, skills, etc.)?
        Did they share preferences that match the persona?

Note: For information not in the persona, the user simulator is expected to make reasonable
     assumptions.
Scale:
        5: Perfect representation, completely authentic
        4: Good representation, minor inconsistencies
        3: Adequate but missing key elements or some inconsistencies
        2: Poor representation, major gaps or contradictions
        1: Very poor, doesn't match persona at all

STEP 3: EVALUATE RECOMMENDER QUALITY
Rate each aspect (1-5):

A. Personal Fit
Question: How well does the final recipe match your individual preferences depicted through
     the persona
Scale:
        5: Excellent adaptation - respects constraints, matches preferences.
         incorporates culture where relevant
        4: Good adaptation - addresses most important aspects well
        3: Adequate adaptation - some awareness but misses key elements
        2: Poor adaptation - generic approach with little personalization
        1: Very poor adaptation - ignored preferences and cultural cues

B. Interaction Quality
Question: How well did they gather relevant information before recommending?
Scale:
        5: Excellent questioning - thorough, relevant, built understanding
        4: Good questioning - adequate information gathering
        3: Adequate questioning - some relevant questions asked
        2: Poor questioning - superficial or irrelevant questions
        1: Very poor questioning - rushed to recommendation without adequate probing

Confidence: How confident are you in your ratings? (High/Medium/Low)
Comment: Any additional thoughts/observations on the conversation, persona alignment, or
    recommendation quality- please share. Anything that stood out, was interesting, or was
     peculiar?
```

**Survey Questions**

**Welcome to the Survey!**

In this task, you will see **9 questions** about different aspects of life in your country.

👉 Your task is to **rank the 9 questions** from **1 to 9**, based on how *common or specific* you think their answers would be across different countries.

- **Rank 1** = Among the 9, this question's answers would be the **most common across countries**.
- **Rank 9** = Among the 9, this question's answers would be the **most specific to your own country/region**, and least likely to be the same elsewhere.
- **Ranks 2–8** = The answers are **somewhere in between**. A lower number (closer to 1) means you think the answers are **more likely to be shared across countries**; a higher number (closer to 9) means the answers are **more country-specific**.

**Examples:**

- Convenient foods like *"overnight oats"* are now popular in many countries → this type of question should be closer to **Rank 1**.
- Currency (e.g., *Indian Rupee, Japanese Yen*) is unique to each country → this type of question should be closer to **Rank 9**.

**Important Guidelines:**

1. The ranking is **relative between the 9 questions**. Even if two questions feel similar, you must decide which one is relatively *more common* and which is *more specific*.
2. If two questions feel similar, pick the one you think is **slightly more common** to give it the higher rank.
3. Use each rank **1–9 exactly once** (no repeats).
4. **This survey contains a few** *attention-check questions*. These are included to make sure participants are carefully following instructions. **If you fail these checks, your responses may be disqualified.**

**An Example with 3 Questions:**
Suppose you are ranking the following **3 sample questions:**

1. Common fast foods in your country
2. National currency
3. National holidays

A possible ranking among the three questions could be:

- Rank 1→ Common fast foods (because many countries share similar fast foods)
- Rank 2→ National holidays (some overlaps, but also country-specific)
- Rank 3→ National currency (unique to each country)

This shows how you should think about the ranking: compare **relative commonness** and use each rank exactly once.

Thank you for your participation!

Figure 7: Survey Instructions

Figure 8: Survey Questions

Figure 9: Attention Checks

