# OpenReview forum: "Measuring Meta-Cultural Competency: A Spectral Framework for LLM Knowledge Structures"
_ICLR.cc/2026/Conference — Submitted to ICLR 2026_

### Official Review · Reviewer_9WWX · 2025-10-27

**Soundness:** 2
**Presentation:** 2
**Contribution:** 2
**Rating:** 2
**Confidence:** 4

**Summary:**

The paper proposes evaluating cultural knowledge in LLMs not as isolated facts but as a broader understanding of relationships across cultures, which the authors term "macrostructures". The approach extracts country-level probability distributions over domain items, builds country×country adjacency matrices (cosine similarity), and summarises those matrices using two spectral metrics: ER and SR. The metric pair is used to categorise domains into four macrostructure types. They evaluate several open models across nine domains and 170 countries and present some interesting findings. They also test these metrics on a recipe-recommendation downstream case study judged by an LLM simulator and a small human validation set. The results claim these spectral metrics capture a distinct, useful notion of “variational awareness” and reveal domain-specific failures.

**Strengths:**

1. To the best of my knowledge, framing cultural evaluation via macrostructures and linking it to CCT theory is a novel cross-disciplinary idea.
2. The framework is formally stated (matrix construction, ER and SR definitions) and yields compact, interpretable summaries (ER = effective dimensionality; SR = λ1/λ2). Makes it easy for other researchers to reproduce and build upon.
3. The finding that macro-structural competency may plateau with scale challenges common assumptions and suggests new evaluation methods.

**Weaknesses:**

1. The core concept of "macrostructures" remains abstract and lacks intuition, particularly in the introduction. The paper would benefit significantly from a clear visualization (e.g., a simplified t-SNE or UMAP of a "good" vs. "bad" country-similarity matrix) to help the reader see what a "high-consensus" (low ER) or "high-diversity" (high ER) structure actually looks like.
2. Section 2.1 seems unnecessary since there is repetition with the introduction. There are no clear definitions, and it is written like a literature review section. The third component of negotiation strategies is not used in the rest of the paper. The section reads like a philosophical justification rather than a rigorous definition section, which leaves the reader with little clarity about how these ideas translate into measurable quantities.
3. L216-L226 provides formal definitions (ER/SR) to measure "variational awareness," but this link feels assumed rather than demonstrated. The connection between, for example, a high Effective Rank and a model's awareness of cultural variation is not explicitly proven. The logic needs to be more thoroughly established.
4. The spectral measures are computed from matrices built using very large, and in some cases scraped, candidate lists (e.g., >3700 items per Table 1). The resulting "macrostructure" could be highly sensitive to the noise, bias, or composition of these lists. The paper needs an ablation study to test this.
5. The method for assigning the four-class labels (LH, HL, etc.) relies on binarizing the ER and SR values using their medians. This threshold is dataset-dependent and potentially arbitrary.
6. Many key claims are presented verbally without sufficient statistical backing. For example, the model rankings (Section 4.2), the performance "plateau" claim, and the correlations in the downstream task (Section 5.2) all require more statistical tests.

**Questions:**

1. Could you provide (or release) the exact candidate lists (CSVs) for each domain and country, and the scraping/cleaning pipeline? This would allow reproducibility and judge whether noisy items influenced ER/SR.
2. To support W4, I recommend the following ablations: compute ER/SR after (i) restricting items to top-k globally frequent items, (ii) removing low-frequency/noisy entries, and (iii) using curated vs. scraped lists to quantify sensitivity. Explicitly report how item selection changed ER/SR for a few representative domains (eg, food, language, currency).
3. To support W5, a few suggestions are (a) show the continuous distributions of ER and SR values so readers can judge separability, (b) test robustness to alternate thresholds (e.g., quartiles or k-means clustering), and (c) use ER and SR as continuous predictors for the downstream task metrics (APR/INT) instead of relying solely on the discretized labels.
4. Please add confidence intervals for the macro-F1 scores, statistical tests (e.g., t-tests) for differences between model pairs, and report effect sizes for the correlations to substantiate these claims. Per-domain confusion matrices would also clarify the evaluation.

---

> ### Author Response · Authors · 2025-11-17
> **Responses Part 1**
>
> We thank the reviewer for acknowledging the novelty of our interdisciplinary framework and are glad they are intrigued by the findings, just as we are. Please see our response to the raised questions below.
>
> ```Weakness 1: The core concept of "macrostructures" remains abstract and lacks intuition, particularly in the introduction...```
>
> We agree that the macrostructure is an inherently abstract concept. In the current draft, we intentionally keep the description brief in the Introduction, with a more detailed and intuitive explanation provided in Section 2.1 (lines 89-99). That said, we fully concur that a concrete visualization would further improve the reader's intuition.
>
> While we already present representative graph structures for each macrostructural category in Figure 1 and describe their characteristics in lines 231-245, we will enhance this by adding an explicit visualization and interpretation in the main paper. The new figure, available at https://figshare.com/s/b8fd9483b8e9dd158014, shows t-SNE representations for all four ER/SR regimes using synthetically generated cosine-similarity matrices (the heatmaps of their adjacency matrices are available here: https://figshare.com/s/7a265cd6dca9683bcc95). Specifically, the bottom-left panel illustrates a near-linear backbone with small perturbations and local patterns (Low ER, High SR), while the top-right panel shows multiple dense clusters representing diverse, high-dimensional structures (High ER, Low SR). The top-left and bottom-right panels depict mixed regimes with varying degrees of consensus and diversity.
>
> For comparison, we will also include the following empirical t-SNE visualizations derived from adjacency matrices across selected domains and all models in the appendix: https://figshare.com/s/0c2eee957491f37f7668. We thank the reviewer again for this excellent suggestion, which will significantly improve the paper’s understandability and intuitive appeal.
>
> ```Weakness 2: Section 2.1 seems unnecessary since there is repetition with the introduction...```
>
> We appreciate the reviewer's perspective and keen observations, and agree that some readers, particularly those already familiar with the underlying disciplines, may prefer a more concise or differently ordered presentation. However, given the inherently interdisciplinary nature of this work, we designed Section 2.1 to serve as an accessible bridge for readers without prior exposure to anthropology or psychology. This section introduces essential concepts such as Cultural Consensus Theory, macro- and micro-structures, and metaculture, which collectively form the basis of the formal framework presented later in Section 3.1.
>
> While an alternative organization, such as presenting the formal framework before the conceptual background, or relocating parts of Section 2.1 to the Discussion, could benefit some readers, we believe the current structure best balances accessibility and rigor, ensuring clarity for a broad and diverse readership.
>
> Also, we included the discussion on negotiation strategy (lines 109-113) solely to provide a complete conceptual picture of meta-cultural competency and do not directly experiment with this component. We will add a clarifying footnote, *“The discussion on negotiation strategy is included only to provide a complete conceptual context for meta-cultural competency and is not experimentally evaluated in this study.”* at line 109 to make this explicit.

---

> > ### Author Response · Authors · 2025-11-17
> > **Responses Part 2**
> >
> > ```Weakness 3: L216-L226 provides formal definitions (ER/SR) to measure "variational awareness," but this link feels assumed rather than demonstrated...```
> >
> > We appreciate the reviewer for highlighting this nuanced and important point. To clarify the connection, we will add a new subsection (Section 3.1.2) immediately following Section 3.1.1, providing the following formal explanation of how ER and SR relate to variational awareness. We believe this addition will make the conceptual link between these measures much clearer to readers.
> >
> > - Using the same notation as in the formal definition (Section 3.1, lines 202-225), let $A^{d_i}$ represent the $n \times n$ country-country similarity matrix for the domain $d_i$.
> > - $A^{d_i} = [s_{ij}] \in R^{n \times n}; s_{ij}$ = cosine similarity($c_i$, $c_j$)
> >
> > Computing the eigendecomposition of $A^{d_i}$,
> >
> > - $A^{d_i} = U \Lambda U^T$; $\Lambda$ = diag ($\lambda_1​,..., \lambda_n$​), where each eigenvalue $\lambda_i$​ represents the strength of a latent pattern of similarity or variation across countries. The eigenvalues capture how the overall structure of the domain’s cross-country relationships is distributed across these latent patterns.
> >
> > - A large first eigenvalue ($\lambda_1$​​) indicates that one dominant pattern explains most similarities (high consensus), whereas a flatter spectrum (multiple comparable eigenvalues) indicates several independent cultural factors (high diversity).
> >
> > - These latent patterns correspond to major axes of cultural differentiation. For example, in the domain of food, one dimension might distinguish rice- versus wheat-based cuisines, another fish- versus meat-dominant diets, and so on, each reflecting a distinct global pattern of variation.
> >
> > - The normalized eigenvalues ($\tilde\lambda_i$​ from Equation 2, lines 223-224) quantify how much each latent cultural pattern contributes to the overall structure across countries.
> >
> > - Thus, the effective rank (ER) measures how many such meaningful patterns exist, capturing diversity or pluralism. The spectral gap ratio (SR) measures the dominance of the leading shared pattern, capturing consensus among countries. Together, the ER and SR define the macrostructure of the domain.
> >
> > By definition (lines 104-105; Saha et al., 2025b), Variational Awareness (VA) is a model’s meta-knowledge of how cultural patterns vary across groups. In spectral terms, this corresponds to recognizing both:
> >
> > - How variation is distributed: reflected by the spectral diversity (ER), which shows how many distinct latent patterns exist.
> > - How consensus is expressed: reflected by the spectral dominance (SR), which shows how strongly a single shared pattern prevails.
> >
> > Thus, formally, VA is the awareness of the spectrum itself. It is the awareness of how many distinct cultural dimensions exist (ER) and how dominant the shared pattern is (SR), where
> >
> > - High VA = the model captures both the diversity of patterns and their relative coherence across countries.
> > - Low VA  = the model misrepresents variation, either by oversimplifying it (homogenization) or fragmenting it (discretization).
> >
> > In summary, VA reflects the model’s meta-knowledge of the spectral composition of cultural patterns - how diversity (ER) and consensus (SR) jointly shape the macrostructure of a domain. We note that this eigenvalue-based formalism is one possible instantiation of VA, consistent with the definition of Saha et al. (2025b) and conceptually grounded in Cultural Consensus Theory (CCT), though other formulations are equally possible.

---

> > > ### Author Response · Authors · 2025-11-17
> > > **Responses Part 3**
> > >
> > > ```Weakness 4: The spectral measures are computed...``` **+** ```Question 2: To support W4, I recommend the following ablations...```
> > >
> > > This is a very interesting and crucial observation. We agree that large candidate lists could, in principle, introduce noise; however, in our case, the lists are already filtered from a far larger universe of possible items. For example, the global diversity of documented and undocumented food items is estimated to range between 16,000 and 100,000 (Abarca, 2016; Lachat, 2018; TasteAtlas, 2024; Wiki), while the total number of unique edible preparations worldwide is plausibly in the millions when local variations are considered. To balance breadth and quality, we restricted our list to ~3,700 food items curated from TasteAtlas, a well-recognized, expert-verified database of regional foods and recipes. This subset is representative yet minimally noisy, and far smaller than the true global inventory, but large enough to avoid collapsing into global stereotypes or overrepresenting only high-frequency "Western" items. In short, the ~3700 items are unlikely to introduce noise; it is the signal of cross-cultural diversity (variational awareness) that our macrostructure-based framework seeks to capture.
> > >
> > > We agree that examining the stability of macrostructures under varying list sizes is an interesting direction for future work, as there is an inherent trade-off to consider. Using too few items would fail to capture the full cultural diversity within each domain (possibly leading to underreporting the ER), while excessively large lists could introduce noise and distort the structure in unpredictable ways (possibly capturing spurious patterns in the noise, leading to misrepresentative ER and SR values). Our chosen item sizes were selected based on domain knowledge to balance coverage and reliability, ensuring sufficient diversity without compromising signal quality, as discussed for the food domain above and in Section A.1 for other domains. Systematically exploring this balance through controlled experiments remains an important area for future investigation.
> > >
> > > ```Weakness 5: The method for assigning the four-class labels (LH, HL, etc.)...``` **+** ```Question 3: To support W5, a few suggestions are...```
> > >
> > > We acknowledge that using the median as a threshold might introduce dataset dependence. However, our choice was intentional and methodological. Our goal was to convert continuous ER and SR values into a classification framework that allows evaluation through interpretable and quantifiable metrics such as the F1 score.
> > >
> > > Conceptually, the underlying relationships between ER, SR, and human-labeled truths remain continuous. In fact, even if the analysis were reformulated using quartile thresholds, k-means clusters, or treated as a regression over continuous values, the relative model rankings and key findings would remain unchanged. The discretization does not alter the spatial arrangement of data points in the ER-SR plane, nor the correspondence to human judgments. It simply provides a consistent and transparent boundary for comparison, without affecting the core findings. We will include the following scatter plot of ER versus SR in the revised version appendix to illustrate these continuous patterns and demonstrate separability: https://figshare.com/s/0030e11fa0c342e5517d
> > >
> > > Also, the human ground-truth labels, collected across 16 world regions and validated by three domain experts (Section 3.2, lines 268-269), are a robust reference irrespective of the thresholding scheme. We also include the human SR distribution for reference in this link: https://figshare.com/s/1fb4e663ac73412604c1, which we will add to the appendix.

---

> ### Author Response · Authors · 2025-11-17
> **Responses Part 4**
>
> ```Weakness 6: Many key claims are presented verbally without sufficient statistical backing...``` **+** ```Question 4: Please add confidence intervals...```
>
> We agree that these tests strengthen the study. Following this feedback, we conducted additional statistical analyses and present the results below.
>
> Since each model’s macro-F1 score is computed from K = 27 observations (9 domains x 3 repetitions), we performed a bootstrap analysis (10,000 resamples) to estimate 95% confidence intervals for the macro-F1 scores. The results are summarized below, and a detailed boxplot is shared here: ​​https://figshare.com/s/3956b85ca02cbd1e5b18.
>
> | model | macro_f1 | low | high |
> |---:|---:|---:|---:|
> | GPT-2 | 0.050 | 0.000 | 0.091 |
> | Llama-3.2-1B-it | 0.183 | 0.068 | 0.278 |
> | GPT-J-6B | 0.215 | 0.083 | 0.319 |
> | Gemma-2-9B-it | 0.333 | 0.161 | 0.483 |
> | Gemma-2-2B-it | 0.349 | 0.171 | 0.495 |
> | Aya-8B | 0.363 | 0.174 | 0.510 |
> | Llama-3.1-70B-it | 0.363 | 0.168 | 0.512 |
> | Llama-3.1-8B-it | 0.374 | 0.195 | 0.522 |
>
> Pairwise t-tests between bootstrapped macro-F1 distributions show that all model pairs, except Aya-8B and Llama-3.1-70B-it, differ significantly (p < 0.05), confirming that the observed ranking differences are statistically meaningful. We also provide the per-domain confusion matrices for all models here: https://figshare.com/s/9c34cf75a923b48cc0f6. Also, please note that the microstructural model rankings mentioned in Section 4.2 (lines 347-348) are as reported by other studies. We do not generate the rankings - we just collate and compare against them.
>
> As suggested by the reviewer, for the downstream recipe recommendation task, we further examined the relationship between macrostructural alignment (Section 4.1) and downstream metrics (APR/INT; Figure 4), and share the new results. We computed Spearman rank correlations between the two ranking sets and observed strong, statistically significant associations: APR: ρ = 0.942 (p < 0.05); INT: ρ = 0.828 (p < 0.05). Bootstrapped 95% confidence intervals (10,000 iterations) were (0.51, 1.0) and (0.03, 1.0), respectively. These results indicate that macrostructural alignment correlates strongly with the recommendation setup - a potential real-world application of culture-specific personalization. However, we are mindful that the recommendation ability is a result of many capabilities, such as explication, variational awareness, and also culture-specific knowledge with complex interactions, multilevel dependencies, and confounders. Independently analyzing the effect of each of these components is a promising future work needing systematic experimentation, which we will point out in the discussion and future work sections
>
> We will add all the above shared details to the appropriate places in the main paper and move the larger illustrations to the appendix.
>
> ```Question 1: Could you provide (or release) the exact candidate lists (CSVs) for each domain and country, and the scraping/cleaning pipeline?...```
>
> Yes, we have made our entire codebase anonymized and publicly available, which includes the full evaluation and analysis pipeline, along with all candidate lists (CSVs) and human evaluation data used for the Macrostructures experiment. The repository can be accessed here: https://anonymous.4open.science/r/MetaCultural-Competence-Macrostructures/.
> We also plan to anonymize and release all human-generated data from the Recipe Recommendation experiment in the final version of the paper to further support transparency and reproducibility.
>
> We sincerely thank the reviewer for taking the time to provide such detailed and thoughtful feedback. Their comments helped us refine several aspects of our work and strengthened the overall study. We deeply appreciate the depth and rigor of their review, especially at a time when thorough, scientific feedback is becoming increasingly rare. We hope the reviewer finds that our revisions address their suggestions to the mark.
>
> **References:**
> 1. Saha, Sougata, Saurabh Kumar Pandey, and Monojit Choudhury. "Meta-Cultural Competence: Climbing the Right Hill of Cultural Awareness." Proceedings of the 2025 Conference of the Nations of the Americas Chapter of the Association for Computational Linguistics: Human Language Technologies (Volume 1: Long Papers). 2025b.
> 2. Wiki: https://en.wikipedia.org/wiki/Lists_of_foods
> 3. Lachat, Carl, et al. "Dietary species richness as a measure of food biodiversity and nutritional quality of diets." Proceedings of the National Academy of Sciences 115.1 (2018): 127-132.
> 4. Abarca, Meredith E. Voices in the kitchen: Views of food and the world from working-class Mexican and Mexican American women. Vol. 9. Texas A&M University Press, 2006.
> 5. TasteAtlas (2024). Global Database of Traditional Foods. Retrieved from https://www.tasteatlas.com.

---

> > ### Author Response · Authors · 2025-11-24
> >
> > Dear Reviewer
> >
> > Once again, we thank you for the thorough and candid assessment. We remain available for any further clarification; as the discussion deadline approaches, we would sincerely appreciate your response.
> >
> > Finally, we request you again to kindly reconsider your scores based on the provided responses.
> >
> > Thank you.

---

> > > ### Author Response · Authors · 2025-11-30
> > > **Summary**
> > >
> > > We sincerely thank the reviewer for their thoughtful and in-depth feedback, and summarize how we addressed each of their concerns and clarified misunderstandings.
> > >
> > > **1. Clarifying macrostructures and improving intuition:** Acknowledging that macrostructures are inherently abstract, we shared new t-SNE visualizations (synthetic and empirical) and adjacency-matrix heatmaps, which will be added to Section 2.1. This directly addresses the reviewer's request and makes the four ER/SR regimes visually concrete.
> > >
> > > **2. Purpose of Section 2.1:** The reviewer questioned redundancy and lack of definitions. We clarified that Section 2.1 is intentionally an interdisciplinary conceptual bridge for readers outside anthropology/psychology. To avoid confusion, we will add a footnote specifying that negotiation strategies are included only for conceptual completeness and are not part of our experiments.
> > >
> > > **3. Formal connection between ER/SR and Variational Awareness:** To correct the misunderstanding that this link was "assumed", we will add a new subsection (Section 3.1.2) that provides a full eigenvalue-based explanation: ER quantifies latent diversity, and SR quantifies consensus. This anchors VA formally and aligns with Cultural Consensus Theory.
> > >
> > > **4. Sensitivity to item-list size:** We clarified that our lists are already curated, filtered, and much smaller than the true domain spaces, making noise unlikely. We discuss the conceptual trade-off and position ablations as important future work that extends beyond the current scope.
> > >
> > > **5. Thresholding of ER/SR:** We clarified that median thresholding is methodological rather than arbitrary and does not affect relative findings. We provided additional ER-SR scatter plots and human SR distributions to illustrate continuity and separability.
> > >
> > > **6. Statistical support:** We added bootstrapped confidence intervals, pairwise significance tests, per-domain confusion matrices, and significance analyses for downstream correlations.
> > >
> > > **7. Reproducibility:** We released the full codebase, candidate lists, and pipelines. We will release the anonymized human evaluation data in the final version of the paper.

---

### Official Review · Reviewer_B5Qz · 2025-10-31

**Soundness:** 3
**Presentation:** 3
**Contribution:** 2
**Rating:** 6
**Confidence:** 2

**Summary:**

this paper solved the problem of evaluating cultural competency in LLMs beyond factual knowledge and surface-level cultural awareness. This paper proposed a spectral-analysis-based framework that measures macrostructural patterns in models’ cultural knowledge—capturing how models internalize and organize cultural variation across domains as an indicator of cultural competency. The framework introduces new spectral metrics to assess cultural pluralism and consensus, validated across eight models and nine domains.

**Strengths:**

1/ This work proposed a novel framework that shifts evaluation from micro-level factual correctness to macro structural cultural understanding. This theoretical move—linking spectral analysis to cultural competency.

2/ The authors conduct comprehensive experiments across models, domains, and human-validated ground truths, demonstrating that the proposed macrostructural evaluation captures distinct aspects of cultural knowledge compared to existing benchmarks.

3/ The simulation-based study connecting macrostructural scores to real-world user alignment (e.g. culturally appropriate recipe recommendations) is a strong practical validation of the framework’s predictive power.

**Weaknesses:**

1/ I think the paper’s methodology could benefit from clearer intuition—the mathematical formalism is ok but may be difficult for non-technical readers to connect to cultural cognition.

2/ The domain selection (nine cultural areas), though diverse, remains limited in scope; I suggest extending to cross-linguistic or non-Western cultural dimensions to test generality.

3/ While the framework is sound, its interpretability could improve. For example, offering case studies showing what specific macrostructural errors look like in practice.

**Questions:**

See weakneses

---

> ### Author Response · Authors · 2025-11-17
> **Responses Part 1**
>
> We thank the reviewer for acknowledging the novelty and comprehensiveness of our interdisciplinary framework. Please see our response to the raised questions below.
>
> ```Weakness 1: I think the paper’s methodology could benefit...```
>
> We appreciate the reviewer's acknowledgement of the mathematical formalism and fully agree that clearer intuition can make the framework more accessible to readers without a technical background. Anticipating this, we already included Section 2.1, which serves as an accessible bridge for readers without prior exposure to anthropology or psychology. This section introduces essential concepts such as Cultural Consensus Theory, macro- and micro-structures, and metaculture, which collectively form the basis of the formal framework presented later in Section 3.1.
>
> In response to this helpful feedback, we will add the following three points:
>
> **Point 1:** A brief explanatory subsection after Section 3.1.1, explicitly connecting cultural cognition and variational awareness to the spectral formulation of macrostructures as below:
>
>
> >- Using the same notation as in the formal definition (Section 3.1, lines 202-225), let $A^{d_i}$ represent the $n \times n$ country-country similarity matrix for the domain $d_i$.
> >- $A^{d_i} = [s_{ij}] \in R^{n \times n}; s_{ij}$ = cosine similarity($c_i$, $c_j$)
> >
> >Computing the eigendecomposition of $A^{d_i}$,
> >
> >- $A^{d_i} = U \Lambda U^T$; $\Lambda$ = diag ($\lambda_1​,..., \lambda_n$​), where each eigenvalue $\lambda_i$​ represents the strength of a latent pattern of similarity or variation across countries. The eigenvalues capture how the overall structure of the domain’s cross-country relationships is distributed across these latent patterns.
> >
> >- A large first eigenvalue ($\lambda_1$​​) indicates that one dominant pattern explains most similarities (high consensus), whereas a flatter spectrum (multiple comparable eigenvalues) indicates several independent cultural factors (high diversity).
> >
> >- These latent patterns correspond to major axes of cultural differentiation. For example, in the domain of food, one dimension might distinguish rice- versus wheat-based cuisines, another fish- versus meat-dominant diets, and so on, each reflecting a distinct global pattern of variation.
> >
> >- The normalized eigenvalues ($\tilde\lambda_i$​ from Equation 2, lines 223-224) quantify how much each latent cultural pattern contributes to the overall structure across countries.
> >
> >- Thus, the effective rank (ER) measures how many such meaningful patterns exist, capturing diversity or pluralism. The spectral gap ratio (SR) measures the dominance of the leading shared pattern, capturing consensus among countries. Together, the ER and SR define the macrostructure of the domain.
> >
> >By definition (lines 104-105; Saha et al., 2025b), Variational Awareness (VA) is a model’s meta-knowledge of how cultural patterns vary across groups. In spectral terms, this corresponds to recognizing both:
> >
> >- How variation is distributed: reflected by the spectral diversity (ER), which shows how many distinct latent patterns exist.
> >- How consensus is expressed: reflected by the spectral dominance (SR), which shows how strongly a single shared pattern prevails.
> >
> >Thus, formally, VA is the awareness of the spectrum itself. It is the awareness of how many distinct cultural dimensions exist (ER) and how dominant the shared pattern is (SR), where
> >
> >- High VA = the model captures both the diversity of patterns and their relative coherence across countries.
> >- Low VA  = the model misrepresents variation, either by oversimplifying it (homogenization) or fragmenting it (discretization).
> >
> >In summary, VA reflects the model’s meta-knowledge of the spectral composition of cultural patterns - how diversity (ER) and consensus (SR) jointly shape the macrostructure of a domain. We note that this eigenvalue-based formalism is one possible instantiation of VA, consistent with the definition of Saha et al. (2025b) and conceptually grounded in Cultural Consensus Theory (CCT), though other formulations are equally possible.
>
> This revised section will illustrate how the Effective Rank (ER) and Spectral Gap Ratio (SR) scores capture the diversity and consensus patterns that define cultural understanding.

---

> > ### Author Response · Authors · 2025-11-17
> > **Responses Part 2**
> >
> > *(Continuation of response to weakness 1)*
> >
> > **Point 2:** We will also include illustrative plots showing what different ER/SR configurations (LL, LH, HL, HH) look like in representational space, further clarifying the intuition behind these measures.
> >
> > >While we already present representative graph structures for each macrostructural category in Figure 1 and describe their characteristics in lines 231-245, we will enhance this by adding an explicit visualization and interpretation in the main paper. The new figure, available at https://figshare.com/s/b8fd9483b8e9dd158014, shows t-SNE representations for all four ER/SR regimes using synthetically generated cosine-similarity matrices (the heatmaps of their adjacency matrices are available here: https://figshare.com/s/7a265cd6dca9683bcc95). Specifically, the bottom-left panel illustrates a near-linear backbone with small perturbations and local patterns (Low ER, High SR), while the top-right panel shows multiple dense clusters representing diverse, high-dimensional structures (High ER, Low SR). The top-left and bottom-right panels depict mixed regimes with varying degrees of consensus and diversity.
> > >
> > >For comparison, we will also include the following empirical t-SNE visualizations derived from adjacency matrices across selected domains and all models in the appendix: https://figshare.com/s/0c2eee957491f37f7668.
> >
> > **Point 3:** We will also add the following scatter plot of model-wise ER versus SR in the appendix to illustrate these continuous patterns and demonstrate separability: https://figshare.com/s/0030e11fa0c342e5517d
> >
> > We are grateful for this suggestion, as incorporating these three points will strengthen the paper’s readability and interdisciplinary appeal without altering its formal rigor.
> >
> > ```Weakness 3: While the framework is sound, its interpretability could improve...```
> >
> > We thank the reviewer for this thoughtful suggestion. To examine the impact of macrostructures in practice, we already conducted a simulation study in Section 5 using recipe recommendation as a case study, which is an interdisciplinary and cross-cultural task requiring integration of knowledge about nutrition, culture, economy, and geography (Abarca, 2006; Khare, 1992; Toledo et al., 2019). We also performed an error analysis, reported in Section 6 (lines 458-469). For instance, our human evaluations showed that while model judgments aligned with GPT-4o as a reference, participants found the recommendations repetitive and shallow, reflecting a collapse of diverse cultural preferences into stereotyped variants - an issue also noted in prior studies on LLMs' cultural reasoning (Shen et al., 2024; Khan et al., 2025).
> >
> > This performance pattern correlates with our findings in Section 4.1, where instruction-tuned LLMs plateau in macrostructural ability beyond approximately 2B parameters, with even 70B models underperforming across all classes. This suggests that scale alone is insufficient and that the absence of robust macrostructural understanding directly limits practical utility and downstream performance in user-facing tasks.
> >
> > That said, the primary goal of this paper is to introduce and validate a formal framework for measuring macrostructural knowledge - an aspect of cultural understanding distinct from factual or microstructural competence. A more detailed qualitative analysis of specific macrostructural error types is indeed an important and promising direction for future work.

---

> > > ### Author Response · Authors · 2025-11-17
> > > **Responses Part 3**
> > >
> > > ```Weakness 2: The domain selection (nine cultural areas), though diverse, remains limited in scope...```
> > >
> > > **Domain Selection:** While we understand the concern regarding scope, our domain selection is in fact designed to comprehensively capture key aspects of cultural variation across societies. As noted in Section 3 (lines 294-296) and summarized in Table 1, our nine domain-specific questions, covering house numbers, religion, food (national, convenient, common, and healthy), holidays, languages, and currency, represent diverse and complementary facets of cultural knowledge.
> > >
> > > According to Newmark's (1988) widely used taxonomy, culture can be classified into five dimensions: material, ecological, social, habitual, and customary. Our domains collectively span all these dimensions and are present in nearly all world cultures, Western and non-Western alike (Katan, 2014). For example, material culture is represented through currency and house numbers, which embody tangible, infrastructural, and economic practices that vary across societies. Ecological culture is reflected through the four food domains (national, convenient, common, and healthy), which capture how geography, climate, and resource availability shape dietary patterns. Social culture is represented by language and religion, which define collective identity and social cohesion. Habitual culture is captured through holidays and daily food practices, which express recurrent, everyday behaviors. Finally, customary culture is embedded within religious and festive traditions, which reflect moral codes, taboos, and shared symbolic values. Moreover, as detailed in Table 2 and Section A.1 (lines 880-886), the item lists for each domain were curated to ensure global coverage, incorporating data from countries across regions and cultures. These considerations were carefully factored into our domain selection but were not explicitly described in the manuscript; we will include them in Section 3.2 for greater clarity.
> > >
> > > Further, as discussed in Section 3.1.1, these nine domains jointly represent all four possible knowledge structures (LL, LH, HL, HH) in our framework, establishing both conceptual and structural coverage. Consequently, the proposed ER and SR metrics are inherently generalizable, as they quantify domain-independent relational patterns - diversity (ER) and consensus (SR) - that apply to any culturally structured system.
> > >
> > > **Language Selection:** We also acknowledge that language is a vital dimension of cultural expression. However, language and culture are not equivalent. In linguistic anthropology, language is viewed as a symbolic resource embedded within broader cultural systems, rather than a direct proxy for culture (Geertz, 1973; Lee, 2016). Our goal in this paper is to assess whether models encode variational knowledge (the structural understanding of cultural diversity), rather than their multilingual proficiency. This is already reflected in our testing setup: although the items are presented in English, they represent culturally diverse content from across the world, allowing us to probe cross-cultural variation independent of language. Moreover, since we evaluate models in English, the language in which they perform best, any absence of rich variational awareness here would almost certainly translate to poorer performance in other languages.
> > >
> > > Recent evidence supports this distinction. Rystrøm et al. (2025) show that stronger multilingual capability does not necessarily yield better cultural alignment in LLMs; models can competently process multiple languages yet still fail to reflect local cultural values. While extending our framework to multilingual settings would indeed be a valuable direction for future work, our present focus is on introducing and validating a white-box framework for measuring knowledge structures. We explicitly acknowledge the absence of multilingual evaluation as a limitation in Section 6 (lines 475-477) and will add the above explanation to make our exposition clearer.
> > >
> > > Nonetheless, we agree that expanding our framework to multilingual and additional domain settings is a valuable next step.

---

> > > > ### Author Response · Authors · 2025-11-17
> > > > **Responses Part 4**
> > > >
> > > > We thank the reviewer again for their excellent suggestions, which will significantly improve the paper’s understandability and intuitive appeal.
> > > >
> > > > **References:**
> > > > 1. Abarca, Meredith E. Voices in the kitchen: Views of food and the world from working-class Mexican and Mexican American women. Vol. 9. Texas A&M University Press, 2006.
> > > > 2. Khare, Ravindra S., ed. The eternal food: Gastronomic ideas and experiences of Hindus and Buddhists. State University of New York Press, 1992.
> > > > 3. Toledo, Raciel Yera, Ahmad A. Alzahrani, and Luis Martinez. "A food recommender system considering nutritional information and user preferences." IEEE Access 7 (2019): 96695-96711.
> > > > 4. Shen, Siqi, et al. "Understanding the Capabilities and Limitations of Large Language Models for Cultural Commonsense." Proceedings of the 2024 Conference of the North American Chapter of the Association for Computational Linguistics: Human Language Technologies (Volume 1: Long Papers). 2024.
> > > > 5. Khan, Ariba, Stephen Casper, and Dylan Hadfield-Menell. "Randomness, not representation: The unreliability of evaluating cultural alignment in llms." Proceedings of the 2025 ACM Conference on Fairness, Accountability, and Transparency. 2025.
> > > > 6. Lee, Ee Lin. "Language and culture." Oxford Research Encyclopedia of Communication. 2016.
> > > > 7. Rystrøm, Jonathan, Hannah Rose Kirk, and Scott Hale. "Multilingual!= multicultural: Evaluating gaps between multilingual capabilities and cultural alignment in llms." arXiv preprint arXiv:2502.16534 (2025).
> > > > 8. Newmark, Peter. A textbook of translation. Vol. 66. New York: Prentice hall, 1988.
> > > > 9. Geertz, C. "Thick Description: Toward an Interpretive Theory of Culture." The Interpretation of Cultures (1973)
> > > > 10. Katan, David. Translating Cultures: An Introduction for Translators, Interpreters and Mediators. Routledge, 2014.
> > > > 11. Saha, Sougata, Saurabh Kumar Pandey, and Monojit Choudhury. "Meta-Cultural Competence: Climbing the Right Hill of Cultural Awareness." Proceedings of the 2025 Conference of the Nations of the Americas Chapter of the Association for Computational Linguistics: Human Language Technologies (Volume 1: Long Papers). 2025b.

---

> > > > > ### Author Response · Authors · 2025-11-24
> > > > >
> > > > > Dear Reviewer
> > > > >
> > > > > Once again, we thank you for recognizing the framework’s novelty and offering constructive suggestions. We hope that our clarifications addressed your questions; as the discussion deadline approaches, we would sincerely appreciate your response.
> > > > >
> > > > > Finally, we request you again to kindly reconsider your scores based on the provided responses.
> > > > >
> > > > > Thank you.

---

> > > > > > ### Author Response · Authors · 2025-11-30
> > > > > > **Summary**
> > > > > >
> > > > > > We are glad the reviewer resonated with our approach. Below is a concise overview of how we addressed their questions.
> > > > > >
> > > > > > **1. Need for clearer intuition behind the formalism:** We provided (i) Intuitive explanations of how ER and SR capture diversity and consensus. (ii) Visualizations of the four macrostructural regimes (LL, LH, HL, HH). (iii) A model-wise ER-SR scatter plot in the appendix. These additions strengthen the paper's accessibility and will be added as a new explanatory subsection after Section 3.1.1.
> > > > > >
> > > > > > **2. Domain selection and scope:** A key misunderstanding was that our nine domains were limited. We clarified that the selected domains span all five dimensions of culture in Newmark's taxonomy (material, ecological, social, habitual, customary) and were curated for global, non-Western coverage. We will add this explanation to Section 3.2.
> > > > > >
> > > > > > Regarding multilinguality, we clarified that our goal is to measure variational awareness, not multilingual proficiency, which we already mentioned as a limitation in the paper. Cultural structure is encoded in the content of items, not in their linguistic surface form. Recent studies (Rystrøm et al., 2025) confirm that multilingual ability does not guarantee cultural alignment. We will add this explanation to make our exposition more explicit.
> > > > > >
> > > > > > **3. Improving interpretability through case studies:** We clarified that Section 5 already includes a realistic cross-cultural case study (recipe recommendation), integrating economics, geography, religion, nutrition, and habit. We added further explanation showing how observed subjective errors (e.g., repetitive or stereotyped recommendations) correspond to macrostructural misalignment. Deeper qualitative case studies are natural future extensions of this work.

---

### Official Review · Reviewer_VQ6R · 2025-10-31

**Soundness:** 2
**Presentation:** 2
**Contribution:** 2
**Rating:** 4
**Confidence:** 5

**Summary:**

This paper introduces a new analysis framework to evaluate LLMs meta-cultural competency, focusing on their ability to understand and navigate cultural diversity beyond factual knowledge. By examining cultural knowledge across 9 domains and 170 countries using two spectral metrics, Effective Rank and Spectral Gap Ratio, the study captures how models represent cultural structures at a macro level. The results show that instruction models align more closely with human cultural patterns, but increasing model size does not always improve performance. A simulation study further demonstrates that these spectral metrics can predict models’ cross-cultural adaptability, offering a novel approach to assessing cultural understanding in LLMs.

**Strengths:**

The paper presents a new spectral analysis framework that measures how LLMs understand cultural structures rather than just recalling facts. The proposed ER and SR can detect the cultural differences and similarity among countries.

It combines ideas from computational modeling, anthropology, and cognitive science to build a novel theoretical foundation for evaluating cultural competence.

The authors test eight models across 9 cultural domains and 170 countries, providing broad and systematic evidence that makes the study both comprehensive and convincing.

**Weaknesses:**

1. The model selection is limited, such as the GPT-2 is too old, Qwen model is representative but missed. Also any of sota close-sourced models is not tested, such as GPT-5, Grok, deepseek, etc. Which make the results and conclusion not solid.

2. No multilingual evaluation, which is quite essential and basic setting in cultural benchmarking works. Language itself is an important dimension of cultural testing.

3. The prompt setting is stable or not, is not proved by the authors. The author should report the variance when rephrasing the prompts, cus model's output logits will be changed.

4. The simulation experiments is only in food domain, which makes it hard to conving that the proposed metrics also works in other cultural tasks.

5. In the main text (line 347), the authors refer to a table located in the appendix. Besides, the formatting of tables and figures is inconsistent: some figures include titles (figure 3) while others (figure 2) do not, and some tables use the three-line style whereas others display full grid lines.

**Questions:**

1. Can the authors provide more theoretical justification or empirical evidence that these spectral properties truly correspond to cultural cognition rather than to distributional variance?

2. Did the authors test prompt consistency quantitatively (e.g., by reporting variance in ER/SR across prompts)? If not, could they comment on potential sensitivity to prompt phrasing?

3. Do the authors expect the ER/SR metrics to generalize to other culturally sensitive tasks such as conversation, education, or healthcare? How to prove it'll also works?

4. The paper finds that model size does not correlate with macrostructural ability. Do the authors have any hypotheses or ablation results explaining why larger models do not necessarily perform better? Is it because logits measurement is not valid setting?

---

> ### Author Response · Authors · 2025-11-17
> **Responses Part 1**
>
> We thank the reviewer for acknowledging the novelty and rigor of our interdisciplinary framework. Please see our response to the raised questions below.
>
> ```Weakness 1: The model selection is limited...```
>
> We thank the reviewer for this thoughtful observation. GPT-2 was intentionally included to illustrate how older models, which are trained on smaller datasets and without instruction or preference tuning, perform within our framework, providing a historical baseline. Our experiments cover eight open-source models of varying sizes (up to 70B parameters), allowing for a diverse and interpretable comparison. We could not include closed-source models such as GPT-5, Grok, or DeepSeek because our framework requires direct access to full model logits, as noted in lines 294-295 and reiterated in Section 6 (lines 472-473). Since these models do not provide such access, they are currently incompatible with our evaluation setup.
>
> ```Weakness 2: No multilingual evaluation...```
>
> We appreciate the reviewer’s comment and agree that language is an important dimension of cultural expression. However, language and culture are not equivalent: in linguistic anthropology, language is treated as a symbolic resource situated within broader cultural systems, not a one-to-one proxy for culture (Geertz, 1973; Lee, 2016).
>
> Our goal in this paper is to assess whether models encode variational knowledge (the structural understanding of cultural diversity), rather than their multilingual proficiency. This is already reflected in our testing setup: although the items are presented in English, they represent culturally diverse content from across the world, allowing us to probe cross-cultural variation independent of language. Moreover, since we evaluate models in English, the language in which they perform best, any absence of rich variational awareness here would almost certainly translate to poorer performance in other languages.
>
> Recent evidence supports this distinction. Rystrøm et al. (2025) show that stronger multilingual capability does not necessarily yield better cultural alignment in LLMs; models can competently process multiple languages yet still fail to reflect local cultural values. While extending our framework to multilingual settings would indeed be a valuable direction for future work, our present focus is on introducing and validating a white-box framework for measuring knowledge structures. We explicitly acknowledge the absence of multilingual evaluation as a limitation in Section 6 (lines 475-477), and will add the above explanation to make our exposition clearer.
>
> ```Weakness 3: The prompt setting is stable or not...``` **+** ```Question 2: Did the authors test prompt consistency...```
>
> As detailed in Section 3.3 (lines 294-298), we already tested the framework’s invariance using three rephrased variants of each prompt (k=3), which are listed in Table 7. As shown in the heatmaps in Figures 5 and 6 (Appendix B.1), the relative performance across domains remains consistent across these variations, indicating that the framework is robust to prompt rephrasing and that the observed results are not sensitive to prompt wording. We further calculate and share the mean, median, and std dev numbers here: https://figshare.com/s/3b4c1525058954dc9f74
>
> Furthermore, we performed additional bootstrap analysis (10,000 resamples) to estimate 95% confidence intervals for the macro-F1 scores in Figure 2. Pairwise t-tests between bootstrapped macro-F1 distributions show that all model pairs, except Aya-8B and Llama-3.1-70B-it, differ significantly (p < 0.05), confirming that the observed ranking differences are statistically meaningful. The 95% confidence intervals are summarized below, and a detailed boxplot is shared here: ​​https://figshare.com/s/3956b85ca02cbd1e5b18.
>
> | model | macro_f1 | low | high |
> |---:|---:|---:|---:|
> | GPT-2 | 0.050 | 0.000 | 0.091 |
> | Llama-3.2-1B-it | 0.183 | 0.068 | 0.278 |
> | GPT-J-6B | 0.215 | 0.083 | 0.319 |
> | Gemma-2-9B-it | 0.333 | 0.161 | 0.483 |
> | Gemma-2-2B-it | 0.349 | 0.171 | 0.495 |
> | Aya-8B | 0.363 | 0.174 | 0.510 |
> | Llama-3.1-70B-it | 0.363 | 0.168 | 0.512 |
> | Llama-3.1-8B-it | 0.374 | 0.195 | 0.522 |
>
> We will add these explanations and details to the paper in Section 4.1.

---

> > ### Author Response · Authors · 2025-11-17
> > **Responses Part 2**
> >
> > ``` Weakness 4: The simulation experiments is only in food domain...``` **+** ```Question 3: Do the authors expect the ER/SR metrics...?```
> >
> > Thank you for bringing up this important point about domain coverage and generalizability. As detailed in Section 3 (lines 294-296) and summarized in Table 1, our empirical analyses already span nine distinct domain-specific questions pertaining to house number, religion, food (national, convenient, common, and healthy), holidays, languages, and currency, where each represents a different aspect of cultural knowledge. For instance, while convenient food and national food both involve food items, they capture different facets of cultural cognition: the former reflects individual and contextual preferences, whereas the latter encodes collective, identity-linked practices.
> >
> > According to Newmark’s (1988) widely used taxonomy, culture can be classified into five dimensions: material, ecological, social, habitual, and customary. Our domains collectively span all these dimensions and are present in nearly all world cultures, Western and non-Western alike (Katan, 2014). For example, material culture is represented through currency and house numbers, which embody tangible, infrastructural, and economic practices that vary across societies. Ecological culture is reflected through the four food domains (national, convenient, common, and healthy), which capture how geography, climate, and resource availability shape dietary patterns. Social culture is represented by language and religion, which define collective identity and social cohesion. Habitual culture is captured through holidays and daily food practices, which express recurrent, everyday behaviors. Finally, customary culture is embedded within religious and festive traditions, which reflect moral codes, taboos, and shared symbolic values. Moreover, as detailed in Table 2 and Section A.1 (lines 880-886), the item lists for each domain were curated to ensure global coverage, incorporating data from countries across regions and cultures. These considerations were carefully factored into our domain selection but were not explicitly described in the manuscript; we will include them in Section 3.2 for greater clarity.
> >
> > Furthermore, as discussed in Section 3.1.1, these nine domains collectively represent all four possible knowledge structures (LL, LH, HL, HH) in our framework, demonstrating its coverage across varying cultural configurations. Consequently, the ER/SR metrics are inherently generalizable, as they capture structural relationships, such as diversity (ER) and consensus (SR), that apply to any domain where such relational patterns exist.
> >
> > Although the downstream simulation in Section 5 uses recipe recommendation as a case study, it is far from being confined to a single domain. Prior research has established that recipe recommendation is an interdisciplinary and cross-cultural task, requiring integration of knowledge about nutrition, culture, economy, and geography (Abarca, 2006; Khare, 1992; Toledo et al., 2019). Dietary practices, for instance, are shaped by religious taboos and moral norms (Rozin, 2005), economic feasibility and ingredient availability (Mintz & Du Bois, 2002), and time-taste trade-offs that differ across societies (Bisogni et al., 2002).
> >
> > Moreover, the recipe recommendation setting engages and tests multiple competencies simultaneously, including personalization, conversational reasoning, cultural sensitivity, and safety awareness (e.g., avoiding hazardous or culturally inappropriate items). To perform well, the model must ask contextually appropriate clarification questions (captured by the INT score) and produce suitable, culturally informed responses (captured by the APR score).
> >
> > Therefore, the recipe recommendation task and the ER/SR metrics inherently draw on multiple cultural domains and operationalize them in a realistic interaction setting, making it a robust and meaningful testbed for evaluating a model’s cultural competency. We will incorporate these additional explanations into Section 5 (lines 359-362) to provide readers with clearer context and intuition.
> >
> > ``` Weakness 5: In the main text (line 347), the authors refer to a table located in the appendix...```
> >
> > We appreciate the reviewer’s attention to detail in noting the formatting inconsistencies. We will standardize the presentation of all tables and figures in the revised version to ensure stylistic uniformity. Regarding Table 5 (referenced in line 347), we intentionally placed it in the Appendix to comply with page limits. If space permits in the final version, we can move it back to the main text for easier reference.

---

> > > ### Author Response · Authors · 2025-11-17
> > > **Responses Part 3**
> > >
> > > ``` Question 1: Can the authors provide more theoretical justification...?```
> > >
> > > We appreciate the reviewer for highlighting this nuanced and important point. To clarify the connection, we will add a new subsection (Section 3.1.2) immediately following Section 3.1.1, providing the following formal explanation of how ER and SR theoretically relate to variational awareness (VA). We believe this addition will make the conceptual link between these measures much clearer to readers.
> > >
> > > - Using the same notation as in the formal definition (Section 3.1, lines 202-225), let $A^{d_i}$ represent the $n \times n$ country-country similarity matrix for the domain $d_i$.
> > > - $A^{d_i} = [s_{ij}] \in R^{n \times n}; s_{ij}$ = cosine similarity($c_i$, $c_j$)
> > >
> > > Computing the eigendecomposition of $A^{d_i}$,
> > >
> > > - $A^{d_i} = U \Lambda U^T$; $\Lambda$ = diag ($\lambda_1​,..., \lambda_n$​), where each eigenvalue $\lambda_i$​ represents the strength of a latent pattern of similarity or variation across countries. The eigenvalues capture how the overall structure of the domain’s cross-country relationships is distributed across these latent patterns.
> > >
> > > - A large first eigenvalue ($\lambda_1$​​) indicates that one dominant pattern explains most similarities (high consensus), whereas a flatter spectrum (multiple comparable eigenvalues) indicates several independent cultural factors (high diversity).
> > >
> > > - These latent patterns correspond to major axes of cultural differentiation. For example, in the domain of food, one dimension might distinguish rice- versus wheat-based cuisines, another fish- versus meat-dominant diets, and so on, each reflecting a distinct global pattern of variation.
> > >
> > > - The normalized eigenvalues ($\tilde\lambda_i$​ from Equation 2, lines 223-224) quantify how much each latent cultural pattern contributes to the overall structure across countries.
> > >
> > > - Thus, the effective rank (ER) measures how many such meaningful patterns exist, capturing diversity or pluralism. The spectral gap ratio (SR) measures the dominance of the leading shared pattern, capturing consensus among countries. Together, the ER and SR define the macrostructure of the domain.
> > >
> > > By definition (lines 104-105; Saha et al., 2025b), Variational Awareness (VA) is a model’s meta-knowledge of how cultural patterns vary across groups. In spectral terms, this corresponds to recognizing both:
> > >
> > > - How variation is distributed: reflected by the spectral diversity (ER), which shows how many distinct latent patterns exist.
> > > - How consensus is expressed: reflected by the spectral dominance (SR), which shows how strongly a single shared pattern prevails.
> > >
> > > Thus, formally, VA is the awareness of the spectrum itself. It is the awareness of how many distinct cultural dimensions exist (ER) and how dominant the shared pattern is (SR), where
> > >
> > > - High VA = the model captures both the diversity of patterns and their relative coherence across countries.
> > > - Low VA  = the model misrepresents variation, either by oversimplifying it (homogenization) or fragmenting it (discretization).
> > >
> > > In summary, VA reflects the model’s meta-knowledge of the spectral composition of cultural patterns - how diversity (ER) and consensus (SR) jointly shape the macrostructure of a domain. We note that this eigenvalue-based formalism is one possible instantiation of VA, consistent with the definition of Saha et al. (2025b) and conceptually grounded in Cultural Consensus Theory (CCT), though other formulations are equally possible.

---

> > > > ### Author Response · Authors · 2025-11-17
> > > > **Responses Part 4**
> > > >
> > > > ```Question 4: The paper finds that model size does not correlate with macrostructural ability..?```
> > > >
> > > > Yes, we found this result particularly interesting and hypothesize that the observed lack of correlation between model size and macrostructural ability arises from two primary factors.
> > > >
> > > > **First**, irrespective of size, most contemporary language models are trained on similar large-scale web datasets and employ comparable training regimes (Touvron et al., 2023; OpenAI, 2023; Villalobos et al., 2024). Web data tends to reflect thin descriptions of culture - outsider or surface-level portrayals emphasizing the unique and exotic - rather than thick descriptions that capture lived, contextualized experiences (Geertz, 1973; Hymes, 2003; Kommers et al., 2025). For instance, food-related web corpora overrepresent “exotic” dishes or stereotypical associations (e.g., pancakes as a typical American breakfast or pizza as an Italian staple), whereas actual daily practices vary significantly across socioeconomic and regional lines. This bias toward thin cultural representations can lead to a plateau in macrostructural knowledge even as models grow larger.
> > > >
> > > > **Second**, macrostructural knowledge requires recognizing causal and relational dependencies among cultural, environmental, and social factors. For example, how rainfall influences agriculture, which in turn shapes dietary traditions and cross-cultural exchanges. Current next-token prediction objectives are not explicitly optimized to capture such relational or causal abstractions (Bender & Koller, 2020; Saha et al., 2025). While larger models often show improved factual recall (microstructural performance), this does not necessarily translate into deeper structural understanding. Our findings, therefore, complement existing cultural benchmarks that focus on content recall by measuring variational awareness (structural awareness) through macrostructural analysis.
> > > >
> > > > The objective of this paper is to establish a formal framework for measuring this complementary kind of knowledge and to empirically validate it. While exploring why model size does not correspond to macrostructural ability is indeed an interesting direction, we leave a detailed investigation of this hypothesis to future work. However, space permitting, we would definitely love to add these speculations to the paper, which we believe can provide interesting directions for future research.
> > > >
> > > > **References:**
> > > > 1. Geertz, C. "Thick Description: Toward an Interpretive Theory of Culture." The Interpretation of Cultures (1973).
> > > > 2. Hymes, Dell. Ethnography, linguistics, narrative inequality: Toward an understanding of voice. Taylor & Francis, 2003.
> > > > 3. Touvron, Hugo, et al. "Llama: Open and efficient foundation language models." arXiv preprint arXiv:2302.13971 (2023).
> > > > 4. Achiam, Josh, et al. "Gpt-4 technical report." arXiv preprint arXiv:2303.08774 (2023).
> > > > 5. Bender, Emily M., and Alexander Koller. "Climbing towards NLU: On meaning, form, and understanding in the age of data." Proceedings of the 58th annual meeting of the association for computational linguistics. 2020.
> > > > 6. Saha, Sougata, and Monojit Choudhury. "User Behavior Prediction as a Generic, Robust, Scalable, and Low-Cost Evaluation Strategy for Estimating Generalization in LLMs." Findings of the Association for Computational Linguistics: ACL 2025.
> > > > 7. Villalobos, Pablo, et al. "Position: Will we run out of data? Limits of LLM scaling based on human-generated data." Forty-first International Conference on Machine Learning. 2024.
> > > > 8. Kommers, Cody, et al. "Meaning Is Not A Metric: Using LLMs to make cultural context legible at scale." arXiv preprint arXiv:2505.23785 (2025)
> > > > 9. Lee, Ee Lin. "Language and culture." Oxford Research Encyclopedia of Communication. 2016.
> > > > 10. Rystrøm, Jonathan, Hannah Rose Kirk, and Scott Hale. "Multilingual!= multicultural: Evaluating gaps between multilingual capabilities and cultural alignment in llms." arXiv preprint arXiv:2502.16534 (2025).
> > > > 11. Saha, Sougata, Saurabh Kumar Pandey, and Monojit Choudhury. "Meta-Cultural Competence: Climbing the Right Hill of Cultural Awareness." Proceedings of the 2025 Conference of the Nations of the Americas Chapter of the Association for Computational Linguistics: Human Language Technologies (Volume 1: Long Papers). 2025b.

---

> > > > > ### Author Response · Authors · 2025-11-24
> > > > >
> > > > > Dear Reviewer,
> > > > >
> > > > > Once again, we thank you for the careful and detailed review and for your time and effort in reviewing our work. We hope that our clarifications resolved your concerns; as the discussion deadline approaches, we would sincerely appreciate your response.
> > > > >
> > > > > Finally, we ask you to kindly reconsider the current rating in light of the discussion.
> > > > >
> > > > > Thank you.

---

> > > > > > ### Author Response · Authors · 2025-11-30
> > > > > > **Summary**
> > > > > >
> > > > > > We are glad the reviewer appreciated the novelty and rigor of our interdisciplinary framework. Below is a concise overview of how we clarified each of their questions.
> > > > > >
> > > > > > **1. Model selection:** We clarified that GPT-2 is included as an intentional historical baseline. Our framework requires full logit access (not available for GPT-5, Grok, DeepSeek, etc.), explaining why closed-source models cannot be evaluated.
> > > > > >
> > > > > > **2. Multilingual evaluation:** We clarified that our goal is to measure variational awareness, which measures structural cultural knowledge, and not multilingual proficiency. We added theoretical grounding from linguistic anthropology and recent empirical findings showing that multilingual strength does not imply cultural alignment. Although we already mentioned multilinguality as a limitation in the paper, we will add these explanations to make the justification more explicit.
> > > > > >
> > > > > > **3. Prompt robustness:** We already evaluated prompt rephrasing invariance (k=3). We added quantitative variance measures and included macro-F1 confidence intervals and significance tests (bootstrapping + pairwise t-tests). These new results will be added to Sections 3.3 and 4.1.
> > > > > >
> > > > > > **4. Domain generalization:** We clarified a misunderstanding: our empirical evaluation already spans nine cultural domains across all dimensions in Newmark’s taxonomy (material, ecological, social, habitual, customary). We expanded Section 3.2 to articulate this coverage and added explanations showing why ER/SR generalize beyond food, including how the downstream recipe task engages multi-domain cultural competencies.
> > > > > >
> > > > > > **5. Formatting issues:** We will standardize all tables/figures and, if space permits, move Table 5 into the main text.
> > > > > >
> > > > > > **6. Theoretical grounding of ER/SR:** We theoretically derived how ER and SR capture diversity and consensus through spectral structure, and their connection with VA. We will add a new subsection (3.1.2) to share this derivation, which proves that these metrics truly reflect spectral properties corresponding to cultural cognition.
> > > > > >
> > > > > > **7. Model-size effects:** We explained two hypotheses regarding why macrostructural ability plateaus with size: lack of “thick” cultural data and lack of causal-relational training objectives.

---

### Official Review · Reviewer_rox3 · 2025-11-09

**Soundness:** 4
**Presentation:** 3
**Contribution:** 3
**Rating:** 8
**Confidence:** 4

**Summary:**

This paper introduces a novel spectral-analysis-based framework for evaluating the meta-cultural competency of Large Language Models (LLMs). It moves beyond traditional, fact-based microstructural cultural evaluations to capture broader macrostructural patterns in cultural knowledge. The authors propose using spectral metrics—Effective Rank (ER) and Spectral Gap Ratio (SR)—to analyze how models organize knowledge across nine cultural domains and 170 countries. They compare eight LLMs of varying sizes and training regimes, finding that instruction-tuned models align better with human macrostructural expectations, but performance does not scale consistently with model size. A simulation-based experiment further demonstrates that macrostructural alignment predicts a model's ability to effectively serve users from unfamiliar cultural backgrounds.

**Strengths:**

Originality & Insight: The application of spectral analysis to cultural macrostructures is a highly original and compelling idea.

Clarity: The distinction between micro- and macro-structures is clearly drawn and well-motivated.

Rigor & Comprehensiveness: The experimental design is thorough, supported by human validation, and covers a wide range of models and cultural domains.

Significance: The findings have important implications for how we evaluate, select, and potentially train LLMs for globally inclusive applications.

**Weaknesses:**

Linguistic & Cultural Generality: The study is limited to English prompts and models. The language-dependence of the identified macrostructures and the framework's generalizability to multilingual or non-English cultural contexts remain open and critical questions.

Domain Coverage: While the nine chosen cultural domains are diverse, they are not exhaustive and may not capture all facets of cultural variation (e.g., social norms, values).

Potential Bias in Simulation: The simulation-based evaluation relies heavily on GPT-4o as a judge. Although validated with human ratings, this still introduces a potential for model-specific bias. Using another strong LLM as an additional judge could further strengthen this part of the argument.

**Questions:**

1.How might your framework be extended to multilingual or non-English cultural settings? Do you expect the macrostructural patterns to be consistent across different prompt languages, and do you have plans for such an investigation?

2.In the simulation experiment, did you consider using an additional, powerful LLM (e.g., Claude, GPT-4) as a second judge to cross-validate the "appropriateness" and "interaction" scores, thereby mitigating potential judge-specific bias?

3.You position macro- and micro-structures as complementary. Have you explored creating a combined metric that integrates your spectral scores (ER/SR) with performance on existing microstructural benchmarks? If so, what were the results?

---

> ### Author Response · Authors · 2025-11-17
> **Responses Part 1**
>
> We thank the reviewer for acknowledging the originality, clarity, rigor, and significance of our interdisciplinary framework. Please see our response to the raised questions below.
>
> ``` Weakness 1: Linguistic & Cultural Generality: The study is limited to English prompts and models... ``` **+** ``` Question 1: How might your framework be extended to multilingual or non-English cultural settings? ... do you have plans for such an investigation?```
>
> We appreciate the reviewer’s comment and agree that language is an important dimension of cultural expression. However, language and culture are not equivalent: in linguistic anthropology, language is treated as a symbolic resource situated within broader cultural systems, not a one-to-one proxy for culture (Geertz, 1973; Lee, 2016).
>
> Our goal in this paper is to assess whether models encode variational knowledge (the structural understanding of cultural diversity), rather than their multilingual proficiency. This is already reflected in our testing setup: although the items are presented in English, they represent culturally diverse content from across the world, allowing us to probe cross-cultural variation independent of language. Moreover, since we evaluate models in English, the language in which they perform best, any absence of rich variational awareness here would almost certainly translate to poorer performance in other languages.
>
> Recent evidence supports this distinction. Rystrøm et al. (2025) show that stronger multilingual capability does not necessarily yield better cultural alignment in LLMs; models can competently process multiple languages yet still fail to reflect local cultural values. While extending our framework to multilingual settings would indeed be a valuable direction for future work, our present focus is on introducing and validating a white-box framework for measuring knowledge structures. We explicitly acknowledge the absence of multilingual evaluation as a limitation in Section 6 (lines 475-477), and will add the above explanation to make our exposition clearer.
>
> ```Weakness 3: Potential Bias in Simulation: The simulation-based evaluation relies heavily on GPT-4o as a judge...``` **+** ```Question 2: In the simulation experiment, did you consider using an additional...?```
>
> We thank the reviewer for this thoughtful comment. As reported in lines 450-454, our human evaluations yielded high agreement with GPT-4o (Spearman's ρ = 0.83 for APR and ρ = 0.89 for INT), indicating strong alignment between human and model-based judgments. Given that human evaluation remains the gold standard for assessing model behavior and cultural reasoning (Lee et al., 2021), we treat these human ratings as the reference baseline. Hence, additional cross-validation with other LLMs was deemed unnecessary.
>
> While we agree that comparing judgments across multiple large models (e.g., Claude or GPT-4) could be an interesting future extension, such validation would likely not alter the central findings about macrostructural differences. Moreover, larger API-based models introduce additional cost constraints that are not negligible. Overall, the high human-LLM correlation strongly suggests that judge-specific bias is minimal, and that GPT-4o's scoring reliably reflects human evaluative patterns in this setting.
>
> ```Question 3: You position macro- and micro-structures as complementary...```
>
> We fully agree that macro- and microstructures are complementary, and that developing an integrated metric combining our spectral measures (ER/SR) with existing microstructural benchmarks would be highly valuable. Our current focus, however, was to formally define and validate macrostructural analysis as a distinct and complementary dimension of model evaluation - one that captures relational and systemic knowledge often overlooked by microstructural metrics, and currently missing in the literature. That said, we see great promise in a holistic framework that unifies both perspectives, enabling clearer insights into when and why models succeed or fail across levels of cultural reasoning. Designing such a composite evaluation, potentially through weighted or correlation-based integration of macro- and micro-level indicators (for example, hierarchical modeling, or taking inspiration from complex-systems theory), is a key direction we intend to pursue in immediate future work.

---

> > ### Author Response · Authors · 2025-11-17
> > **Responses Part 2**
> >
> > ``` Weakness 2: Domain Coverage: While the nine chosen cultural domains are diverse...```
> >
> > While we understand the concern that our domain selection might come across as limited, it is in fact designed to comprehensively capture key aspects of cultural variation across societies. As noted in Section 3 (lines 294-296) and summarized in Table 1, our nine domain-specific questions, covering house numbers, religion, food (national, convenient, common, and healthy), holidays, languages, and currency, represent diverse and complementary facets of cultural knowledge.
> >
> > According to Newmark’s (1988) widely used taxonomy, culture can be classified into five dimensions: material, ecological, social, habitual, and customary. Our domains collectively span all these dimensions and are present in nearly all world cultures, Western and non-Western alike (Katan, 2014). For example, material culture is represented through currency and house numbers, which embody tangible, infrastructural, and economic practices that vary across societies. Ecological culture is reflected through the four food domains (national, convenient, common, and healthy), which capture how geography, climate, and resource availability shape dietary patterns. Social culture is represented by language and religion, which define collective identity and social cohesion. Habitual culture is captured through holidays and daily food practices, which express recurrent, everyday behaviors. Finally, customary culture is embedded within religious and festive traditions, which reflect moral codes, taboos, and shared symbolic values. Moreover, as detailed in Table 2 and Section A.1 (lines 880-886), the item lists for each domain were curated to ensure global coverage, incorporating data from countries across regions and cultures. These considerations were carefully factored into our domain selection but were not explicitly described in the manuscript; we will include them in Section 3.2 for greater clarity.
> >
> > Further, as discussed in Section 3.1.1, these nine domains jointly represent all four possible knowledge structures (LL, LH, HL, HH) in our framework, establishing both conceptual and structural coverage. Consequently, the proposed ER and SR metrics are inherently generalizable, as they quantify domain-independent relational patterns - diversity (ER) and consensus (SR) - that apply to any culturally structured system.
> >
> > Finally, the downstream recipe-recommendation simulation in Section 5 further demonstrates generalizability. This task is inherently interdisciplinary and cross-cultural, drawing on knowledge of nutrition, culture, economy, and geography (Abarca, 2006; Khare, 1992; Toledo et al., 2019) and engaging competencies such as personalization, conversational reasoning, cultural sensitivity, and safety awareness. Together, the domain-specific evaluations and simulation experiments provide a comprehensive and realistic testbed for assessing a model’s cultural competency, encompassing most major facets of cultural variation. Nonetheless, we agree that a larger-scale future study spanning additional domains would likely reveal further domain-specific patterns. However, the core insight of this work remains unchanged: macrostructural analysis captures complementary information about models - information that current evaluation approaches overlook. We will incorporate these additional explanations into Section 5 (lines 359-362) to provide readers with clearer context and intuition.
> >
> > **References:**
> > 1. Lee, Ee Lin. "Language and culture." Oxford Research Encyclopedia of Communication. 2016.
> > 2. Rystrøm, Jonathan, Hannah Rose Kirk, and Scott Hale. "Multilingual!= multicultural: Evaluating gaps between multilingual capabilities and cultural alignment in llms." arXiv preprint arXiv:2502.16534 (2025).
> > 3. Newmark, Peter. A textbook of translation. Vol. 66. New York: Prentice hall, 1988.
> > 4. Geertz, C. "Thick Description: Toward an Interpretive Theory of Culture." The Interpretation of Cultures (1973)
> > 5. Katan, David. Translating Cultures: An Introduction for Translators, Interpreters and Mediators. Routledge, 2014.
> > 6. Van der Lee, Chris, et al. "Human evaluation of automatically generated text: Current trends and best practice guidelines." Computer Speech & Language 67 (2021): 101151.
> > 7. Abarca, Meredith E. Voices in the kitchen: Views of food and the world from working-class Mexican and Mexican American women. Vol. 9. Texas A&M University Press, 2006.
> > 8. Khare, Ravindra S., ed. The eternal food: Gastronomic ideas and experiences of Hindus and Buddhists. State University of New York Press, 1992.
> > 9. Toledo, Raciel Yera, Ahmad A. Alzahrani, and Luis Martinez. "A food recommender system considering nutritional information and user preferences." IEEE Access 7 (2019): 96695-96711.

---

> > > ### Author Response · Authors · 2025-11-24
> > >
> > > Dear Reviewer,
> > >
> > > Once again, we thank you for the thoughtful and encouraging evaluation and for your time and effort in reviewing our work. We hope that our clarifications addressed any remaining points; as the discussion deadline approaches, we would sincerely appreciate your response.
> > >
> > > Finally, we kindly ask you to reconsider your score in light of the discussion.
> > >
> > > Thank you.

---

> > > > ### Author Response · Authors · 2025-11-30
> > > > **Summary**
> > > >
> > > > We are glad the reviewer positively aligned with our approach. Below is a concise overview of how we addressed their questions.
> > > >
> > > > **1. Linguistic & cultural generality:** We clarified a core misunderstanding that our goal is to measure variational awareness, not multilingual proficiency, which we already mentioned as a limitation in the paper. Language is a symbolic medium, not a proxy for culture. Although prompts are in English, items span worldwide cultural content, allowing cross-cultural macrostructures to emerge independent of language. Recent studies (Rystrøm et al., 2025) confirm that multilingual ability does not guarantee cultural alignment. We will add these explanations to clarify things in the paper.
> > > >
> > > > **2. Domain coverage:** We clarified that the nine domains are not limited. In fact, they intentionally span all five dimensions of culture in Newmark’s taxonomy: material, ecological, social, habitual, and customary, with globally curated item lists. These domains also cover all four macrostructural regimes (LL/LH/HL/HH). The downstream recipe simulation further tests cross-domain cultural reasoning. We will add these explanations to Sections 3.2 and 5.
> > > >
> > > > **3. Potential judge bias (simulation):** We clarified that GPT-4o was validated against human judgments, with high correlations (ρ = 0.83/0.89). Given that human evaluation is the gold standard, additional LLM judges were unnecessary.
> > > >
> > > > **4. Combined macro-micro metrics:** Our present contribution focuses on introducing the framework first, by defining and validating macrostructural analysis as a distinct, complementary dimension. We agree on the value of integrating ER/SR with microstructural benchmarks and identify this as immediate future work.

---

### Meta-Review · Area_Chair_XRLD · 2026-01-08

**Summary:**

While reviewers acknowledged the novelty of applying spectral analysis to evaluate cultural macrostructures, the consensus leans towards rejection due to significant concerns regarding the theoretical validity of the metrics, the robustness of the experimental design, and the limited scope of the evaluation. After checking the authors' response, I think Reviewer 9WWX may increase the final score. I am recommending a rejection, but I wouldn't mind if the paper gets accepted.

**Reviewer Concerns:**

1) Theoretical Validity: Multiple reviewers (9WWX, VQ6R, B5Qz) questioned the validity of the core premise. They argued that the connection between the proposed spectral metrics (Effective Rank and Spectral Gap Ratio) and the abstract concept of "variational awareness" or "cultural competency" is assumed rather than rigorously demonstrated. Reviewer 9WWX noted that the link feels "philosophical" rather than empirically proven. The authors provide a reference  (lines 104-105; Saha et al., 2025b) to prove that Variational Awareness (VA) is a model’s meta-knowledge of how cultural patterns vary across groups. However, since this reference was recently published and not been verified on a large scale.

2) Arbitrary Metrics: Reviewer 9WWX raised critical concerns about the robustness of the metric calculation, noting that the "macrostructure" results could be highly sensitive to noise in the large, scraped candidate lists ($>3700$ items). The authors agree that examining the stability of macrostructures under varying list sizes is an interesting direction for future work, as there is an inherent trade-off to consider.

3) Statistical Significance: Claims regarding performance plateaus and model rankings were presented without sufficient statistical backing, such as error bars or significance tests. The authors provide sufficient statistical information in the rebuttal. However, they failed to add the explanation into the pdf submission during the rebuttal phase.

**Reviewer Scores:**

Reviewer rox3: Maintain the score as 8

Reviewer VQ6R: Maintain the score as 4

Reviewer B5Qz:  Maintain the score as 6

Reviewer 9WWX: Increase the score to 4

---

### Decision · Program_Chairs · 2026-01-26

Reject